# Southern Annular Mode Persistence and Westerly Jet: A Reassessment Using High-Resolution Global Models

Ting-Chen Chen<sup>1</sup>, Hugues Goosse<sup>1</sup>, Cécile Davrinche<sup>1</sup>, Stephy Libera<sup>1</sup>, Christopher Roberts<sup>2</sup>, Matthias Aengenheyster<sup>2</sup>, Kristian Strommen<sup>3</sup>, Malcolm Roberts<sup>4</sup>, Rohit Ghosh<sup>5</sup>, Jin-Song von Storch<sup>6</sup>

- 5 <sup>1</sup>Earth and Life Institute, Université Catholique de Louvain, Louvain-la-Neuve, Belgium
  - <sup>2</sup>European Centre for Medium Range Weather Forecasting (ECMWF), Reading, United Kingdom
  - <sup>3</sup>Department of Atmospheric, Oceanic, and Planetary Physics, University of Oxford, Oxford, UK
  - <sup>4</sup>Met Office Hadley Centre, Exeter, United Kingdom
  - <sup>5</sup>Alfred Wegener Institute Helmholtz Centre for Polar and Marine Research, Bremerhaven, Germany
- 10 <sup>6</sup>Max-Planck Institute for Meteorology, Hamburg, Germany

Correspondence to: Stephy Libera (stephy.libera@uclouvain.be)

**Abstract.** This study evaluates the performance of high-resolution (grid sizes of 9–28 km for the atmosphere; 5–13 km for the ocean) global simulations from the EERIE project in representing the persistence of the Southern Annular Mode (SAM), a leading mode of Southern Hemisphere climate variability. Using the decorrelation timescale of the SAM index ( $\tau$ ), we compare EERIE simulations with CMIP6 models and ERA5 reanalysis.

EERIE simulations reduce long-standing biases in SAM persistence, especially in early summer, with  $\tau$  values of 9–20 days compared to CMIP6's 9–32 days and ERA5's 11 days. This improvement correlates with a more accurate climatological jet latitude ( $\lambda_0$ ). EERIE atmosphere-only AMIP runs outperform the coupled simulations in both  $\tau$  and  $\lambda_0$ , showing smaller biases and ranges of variability, underscoring the critical role of SST representation in shaping atmospheric circulation. In these AMIP experiments, the atmospheric eddy feedback strength, combined with the damping timescale estimated via friction, correlates more strongly with  $\tau$  than  $\lambda_0$ . We speculate that the well-captured jet position (biases <1° relative to ERA5), due to prescribed SSTs, limits  $\lambda_0$ 's explanatory power for  $\tau$  differences, allowing other processes to dominate. Using a finer model grid (9 km vs. 28 km) of the same AMIP model reduces  $\tau$ , though the mechanism remains unclear. Finally, motivated by the importance of oceanic eddies in the Southern Ocean, we conducted sensitivity experiments that filter transient mesoscale features from the

SST boundary conditions. The results suggest that oceanic eddies may enhance summertime SAM persistence (by ~2 days), though this signal is not statistically significant and is absent in the single 9-km run, pointing to a subtle role of mesoscale ocean-atmosphere interaction that remains to be explored.

#### 1 Introduction

40

45

Over the extratropical Southern Hemisphere, the daily- to decadal climate variability is dominated by the Southern Annular Mode (SAM), a mode of natural variability manifested in the large-scale oscillation of atmospheric mass between mid- and high-latitudes and hence changes of the eddy-driven jet in the midlatitudes (e.g., Fogt and Marshall, 2020). This internal variability both influences and is influenced by the atmospheric circulation, affecting regional temperatures and precipitation patterns, sea ice extent, and ocean circulation, with consequences for global heat and carbon redistribution (e.g., Doddridge and Marshall, 2017; Gillett et al., 2006; Lefebvre and Goosse, 2005; Lenton and Matear, 2007; Lovenduski and Gruber, 2005).

As implied by its name "annular", the spatial structure of SAM is approximately "ring-shaped" when viewed from above the South Pole and is nearly barotropic in the vertical direction (Gerber et al., 2010). During the positive phase of SAM, lower air pressure anomalies overlay Antarctica while higher pressure anomalies spread over the mid-latitudes, and such anomalous pressure distribution indicates a strengthening and poleward shifting of the westerly jet that climatologically sits at around 50°S (Lim et al., 2013). While the SAM can, to a first approximation, be described from a zonal-mean perspective, its structure can deviate from the zonal mean and vary across different timescales, affected by factors such as the seasonal cycle of midlatitude jet (atmospheric eddy activity), sea surface temperature (SST) variability, tropical oscillations such as the El Niño-Southern Oscillation (ENSO), stratosphere-troposphere interactions and so on (e.g., Campitelli et al., 2022; Ding et al., 2012; Fogt and Marshall, 2020; Karoly, 1989). On the seasonal scale, SAM is overall more zonally symmetric in austral summer (DJF) but exhibits asymmetric wavenumber 3 components when entering autumn (MAM) and winter (JJA). Readers interested in a comprehensive review of the SAM literature are encouraged to consult Fogt and Marshall (2020) and Thompson et al. (2011).

A key characteristic of SAM is its temporal persistence, referring to how long a given phase of the SAM (positive, negative or neutral) tends to last before transitioning. This long persistence is important as it provides a source of predictability at a timescale longer than the one associated with synoptic variability (e.g., Robinson 2000; Lorenz and Hartmann 2001, Simpson and Polvani 2016). SAM persistence is often measured as the decorrelation timescale (e-folding timescale) which indicates the average duration over which the SAM index remains strongly correlated with its past values. A standard explanation attributes the extended SAM persistence to the reinforcement of westerly flow anomalies by atmospheric eddy momentum fluxes, which are generated by changes in the mean flow and counteract dissipation from surface friction. Several mechanisms may contribute to the eddy-mean flow feedback that reinforces the shifted jet. These include barotropic processes, such as anomalous wave propagation and breaking, and baroclinic processes related to enhanced eddy generation and increased lower-tropospheric baroclinicity in response to shifts in the westerly winds (e.g., Robinson 2000, Lorenz and Hartmann 2001, Zurita-Gotor et al. 2014, Hassanzadeh and Kuang, 2019). Westerly flow anomalies also induce changes in the diabatic heating and cooling—through latent heat release and cloud radiative effects—which alter temperature gradients and, in turn, affect SAM persistence (Xia and Chang 2014, Smith et al. 2024, Vishny et al. 2024). In addition to this eddy-mean flow feedback, SAM persistence can have an origin from the stratosphere, which introduces some non-stationary forcing to SAM. The main influence is likely in late spring and summer at the time of the seasonal breakdown of the stratospheric vortex (Simpson et al. 2011, Byrne et al. 2016, Byrne et al. 2017, Saggioro and Shepherd 2019). Furthermore, interactions between a stationary

While global climate models (GCMs) have shown good skill in capturing the spatial structure of SAM variability, a long-standing challenge for GCMs is that they tend to overestimate the SAM persistence during the austral summer. Based on global reanalysis data, the SAM decorrelation timescale is found to be approximately 10 days on annual mean and is a couple of days higher in early summer (November–January; NDJ), during which period GCMs typically show values that are two to three-times larger (Bracegirdle et al., 2020). Overly persistent SAM in GCMs is correlated with a common bias in the climatological jet position, whereby the simulated tropospheric jets are placed too far equatorward (e.g.,

mode and a propagating mode of the zonal variability could also affect SAM persistence (Lubis and

Hassanzadeh 2021, Sheshadri and Plumb 2017, Smith et al. 2024).

Kidston and Gerber, 2010; Simpson et al., 2013a, b; Simpson and Polvani, 2016; Son et al., 2010). A possible explanation is that models with lower latitude jets exert stronger eddy-mean flow feedback to maintain SAM (Codron, 2005; Simpson and Polvani, 2016).

However, the climatological position of the midlatitude jet is not the only factor for the overly persistent SAM variability in GCMs. Simpson et al. (2013a) performed a series of experiments with nudging and bias correcting procedures using a stratosphere-resolving GCM, the Canadian Middle Atmosphere Model (CMAM). They found that the SAM persistence bias remains even when the representation of the climatological tropospheric winds is artificially improved. Similar conclusions are obtained when another common bias for the overly-persistent summertime SAM —the delayed breakdown of the stratospheric vortex— was manually nudged toward the reanalysis-based seasonal climatology. These results suggest that they may not be the only underlying causes of the SAM persistence bias.

As GCMs improve in their representation of physics, resolution, and overall complexity, some advancements have been made in reducing biases associated with SAM persistence and the climatological jet latitude. Compared to earlier versions of Coupled Model Intercomparison Project (CMIP) models, noticeable reductions in these biases have been reported. Bracegirdle et al. (2020) found that the ensemble-mean bias in the westerly jet latitude decreased from 1.9° in CMIP5 to 0.4° in CMIP6 on an annual mean basis. Consistently, the early-summertime SAM persistence was reduced from approximately 30 days in CMIP5 to 20 days in CMIP6. Nevertheless, the SAM decorrelation timescale remains systematically biased. While higher resolution is generally regarded as beneficial, it is worth exploring whether additional improvements are achievable by further increasing the resolution or if other factors become increasingly significant when the resolution is sufficiently high.

100

Here we revisit this issue using new high-resolution simulations from the Horizon Europe project European Eddy-Rich Earth System Models (EERIE) (M. J. Roberts et al., 2024a). A distinctive feature of the atmosphere-ocean coupled Earth System Models (ESMs) built under EERIE is their adoption of high oceanic resolutions (grid size of 5–13 km) to explicitly represent ocean mesoscale processes, which have been increasingly recognized as critical for weather and climate simulation (e.g., Busecke and Abernathey, 2019; Chassignet and Xu, 2021; Hewitt et al., 2020). Mesoscale oceanic features can influence SAM persistence by strongly affecting surface heat fluxes and surface stress in the Southern

Ocean—a hotspot of mesoscale activity (Frenger et al., 2013; Bishop et al., 2017). These ocean-atmosphere interactions can alter atmospheric temperature gradients and boundary layer structure, modifying diabatic heating and low-level baroclinicity, both of which have been linked to SAM persistence (Xia and Chang, 2014; Smith et al., 2024; Robinson, 2000; Zurita-Gotor et al., 2014). Furthermore, surface stress also plays a role as it tends to damp the westerly winds but also to enhance baroclinicity and the baroclinic feedback (Robinson 2000, Zurita-Gotor 2014, Vishny et al. 2024). EERIE also includes a suite of atmosphere-only simulations and idealized experiments to facilitate exploration of the atmosphere response to the ocean mesoscales by excluding effects attributed to the air-sea coupling and SST biases. Those experiments will allow disentangling the role of the explicit resolution of the eddies compared to the one of increasing the model resolution. Using those experiments, we specifically analyze the potential role of the mesoscale oceanic eddies on SAM persistence, a contribution that has been studied to a minimum to date. The data sources and diagnostics are detailed in Sections 2

and 3, respectively, followed by the results in Section 4 and the conclusions in Section 5.

#### 125 **2 Data**

130

115

120

#### 2.1 EERIE models & simulations

Running from January 2023 to December 2026, the EERIE project aims to build new generations of ESMs run at "eddy-rich resolution" (note that the "eddy" here refers to ocean eddies), which explicitly resolve ocean mesoscale processes with scales of 10–100 km. Crucial components at this scale include mesoscale eddies (analogous to cyclones in the atmosphere) and boundary/frontal currents. EERIE will deliver simulations over multi-centennial timescales centered on four global coupled ESMs and two atmosphere-only models, with an overarching objective to reveal and to quantify the role of ocean mesoscales in shaping the climate trajectory over seasonal to centennial time scales, regionally and globally (European Commission, 2022).

#### 135 **2.1.1** Coupled simulations

This study evaluates the preliminary EERIE Phase 1 simulations (Wachsmann et al., 2024). To facilitate direct comparison across experiments, all outputs were regridded to a uniform 0.25° ×0.25° grid prior to analysis, except for the westerly jet location identification (Section 3.2). A detailed description of the EERIE models can be found in M. J. Roberts et al., (2024a), and Table 1 briefly summarizes the simulations used in the current study. IFS-FESOM2 and ICON model simulations are being conducted 140 following a protocol similar to the CMIP6 HighResMIP (High Resolution Model Intercomparison Project; Haarsma et al., 2016). The HadGEM3-GC5-EERIE model simulation follows protocol similar to CMIP6 DECK (Diagnostic, Evaluation and Characterization of Klima; Eyring et al., 2016). HighResMIP differs from CMIP6 DECK primarily in its use of 1950s' climate conditions instead of 1850s' as the initial 145 state and a shorter spin-up (~50 years instead of >= 200 years; which have been discarded and not counted in the simulation length shown in Table 1) due to the computational demands of high-resolution models. Using the IFS-FESOM2 model, we analyze a 65-year control simulation conducted under fixed 1950 forcings (referred to as 1950control), along with a historical simulation covering the period from 1950 to 2014. For the ICON model, the 22-year 1950control and the historical run are analyzed. For the 150 HadGEM3 model, we examine a 30-year pre-industrial control simulation (piControl) forced by 1850 conditions.

As these simulations cover different time periods and some of them include transient forcing, linear and low-frequency nonlinear trends are removed as standard procedures in the SAM-related diagnostics. This should reduce the impact of the difference in experimental design on the evaluation of the model performance. However, this removal does not fully eliminate the non-stationary features that could have a clear influence on the evaluation of SAM persistence and of the eddy feedbacks (Byrne et al. 2016). We therefore adopt a bootstrapping procedure (Section 3.1) to provide partial quantification of the influence of non-stationarity and uncertainty due to the short period of some simulations. As will be shown later, through bootstrapping resampling, different results can be obtained with the same model even after detrending. We also provide results using two different periods of the ERA5 reanalysis (Section 2.2) as references for comparisons. Note that the difference can be partly attributed to the larger data coverage after 1979 in ERA5.

Table 1. EERIE simulations analyzed in the current study.

| Institution                            | Alfred Wegener Institute (AWI)                 | Max Planck Institute (MPI-M)                     | Met Office (MO)         | European Centre for<br>Medium-range Weather<br>Forecasting (ECMWF) |
|----------------------------------------|------------------------------------------------|--------------------------------------------------|-------------------------|--------------------------------------------------------------------|
|                                        | Coupled atmosphere-ocean models (eddy-rich)    |                                                  |                         | Atmospheric model                                                  |
| System name                            | IFS-FESOM2                                     | ICON                                             | HadGEM3-GC5-<br>EERIE   | IFS                                                                |
| Model                                  | IFS CY48R1,                                    | ICON-A,                                          | UM,                     | IFS CY48R1                                                         |
| components                             | FESOM2, FESIM2                                 | ICON-O                                           | NEMO4.0.4, SI3          |                                                                    |
| Atmos. grid (km)                       | Tco1279 (~9 km)                                | R2B8 (~10 km)                                    | N640 (~20 km<br>at 50N) | Tco1279 (~9 km) Tco399 (~28 km)* *five ensemble members            |
| Atmos. vertical levels (model top)     | 137 (0.01 hPa)                                 | 90 (0.01 hPa)                                    | 85 (85 km)              | 137 (0.01 hPa)                                                     |
| Ocean grid (km)                        | NG5 (~13-5 km)                                 | R2B9 (~5 km)                                     | eORCA12 (~8 km)         | -                                                                  |
| Ocean vertical levels                  | 70                                             | 72                                               | 75                      | -                                                                  |
| Protocol                               | CMIP6 HighResMIP                               |                                                  | CMIP6 DECK              | HighResMIP2                                                        |
| Simulations (analyzed segment lengths) | 1950control (65 yrs)<br>Historical (1950–2014) | 1950control (22 years)<br>Historical (1950-2014) | piControl<br>(30 yrs)   | Historical (1980–2023)                                             |

## 165 2.1.2 Atmosphere-only simulations & sensitivity experiments

The EERIE AMIP simulations were performed for the historical period of 1980–2023 following the HighResMIP2 highresSST-present experimental design (M. J. Roberts et al., 2024b). We analyze the simulations produced with the IFS model in two model grid sizes (~28 km and ~9 km; both with convection parameterization), and the higher-resolution configuration is identical to the atmosphere component of the coupled IFS-FESOM2 (Table 1). One member has been performed at the 9-km resolution, but the 28-km simulations are supplemented with five ensemble members to represent a range of model uncertainty or noise. These ensembles are generated by perturbing the atmospheric initial

conditions for January 1, 1980, using the same methodology employed in operational ECMWF ensemble forecasts (C. Roberts et al., 2024a).

The prescribed boundary conditions are taken from the daily-mean SST reanalysis from the European Space Agency Sea Surface Temperature Climate Change Initiative (ESA CCI SST v3) and the daily-mean sea-ice concentration from the European Organisation for the Exploitation of Meteorological Satellites (EUMETSAT) Ocean and Sea Ice Satellite Application Facility (OSI-SAF), both retrieved on a 0.05° ×0.05° grid. External radiative forcings are generally specified following CMIP6/HighResMIP protocols, and the specificity can be found in C. Roberts et al. (2024a).

To explore the atmospheric response to extratropical ocean mesoscale features, EERIE project also conducted idealized experiments with modified SST boundary conditions. Taking the IFS-AMIP simulations as the control experiments (denoted as ObsSST), NoEddies experiments have the transient oceanic eddy features removed from their SST boundary conditions with a spatial low-pass filter applied to the SST anomaly field (difference from the climatological mean). Sea ice cover remains unchanged in NoEddies. We emphasize that such a design only allows us to test ocean eddies' direct thermodynamic impact (as reflected in SSTs) but not their mechanical influence (through the so-called wind stress feedback or relative winds-currents effects).

The employed filter is a Gaussian filter from the GCM-Filters Python package (Loose et al., 2022). The filter length scale is set to be  $20L_R$ , where  $L_R$  is the spatially varying, climatological Rossby radius in the ocean with a lower and higher limit of 30 km and 700 km, respectively. The filter with a smaller  $L_R$  at high latitudes effectively removes the smaller oceanic eddies there. However, it also removes the larger-scale tropical instability waves near the equator when  $L_R$  reaches its maximum. This potentially obscures the impact of targeted extratropical ocean mesoscales due to tropical-extratropical teleconnections. To avoid this, low-latitude areas are masked out from the filtering with a function ranging from 0 to 1:  $M(\lambda) = \frac{1}{2} \left( \tanh \left( \frac{|h-\lambda|}{s} \right) + 1 \right)$ , where h=10 determines the latitude where the M value is halved (0.5) and s=3 scales the steepness of the masking function. Like the ObsSST, the NoEddies experiment is run with two model grid sizes of ~28 km (five ensemble members) and ~9 km (one member). For more details of the experimental design, we refer readers to C. Roberts et al. (2024b).

#### 200 2.2 CMIP6 models & ERA5 reanalysis

For the diagnostics of SAM persistence and westerly jet characteristics, the CMIP6 models are used to compare with EERIE models. We analyze 31 CMIP6 historical simulations from the first ensemble member that provide outputs of daily geopotential at 500-hPa level and monthly zonal wind at 850 hPa. All CMIP6 outputs are regridded to a uniform 1°×1° grid with the bilinear interpolation before performing the analysis and only the period of 1980-2014 is extracted to ensure a uniform data length. As a proxy of observation, we use the global reanalysis dataset ERA5 (Hersbach et al., 2020) for the same variables and a total period from 1958 to 2023 to cover the earlier period included in some EERIE simulations. Among the reanalysis products that extend backwards in time beyond 1979 (ERA5, 20CRv3, JRA-55), ERA5 is found to agree best with station observations and produces good representation of SAM, both before and after the advent of satellite sounder data (Marshall et al., 2022). While we analyze ERA5 on the commonly distributed 0.25° ×0.25° grid, we have tested the impact with regridding it to the 1°×1° grid and found no notable changes in our results.

## 3 Diagnostics

For the overall assessment of model performance, the diagnostics described in subsections 3.1 and 3.2 below are applied to all available CMIP6 historical and EERIE simulations. Due to the limited accessibility of the EERIE data at the time of writing, diagnostics in subsections 3.3 and 3.4 are only performed on the EERIE atmosphere-only sensitivity experiments to provide deeper investigation on the tropospheric mechanisms critical to the SAM persistence.

## 3.1 SAM persistence timescale

- Some variations exist in the definition of the SAM across the literature (Ho et al., 2012), and its persistence estimation may be sensitive to the methods employed. While many studies adopt similar methodological concepts, the details are often not fully transparent. To ensure clarity, we provide a step-by-step explanation of our approach. Note that SAM is a rather barotropic feature, so even though some traditional definitions consider the vertical averaged field, we have chosen to follow Bracegirdle et al.
- (2020) using a single level for simplicity.

We define the SAM as the first empirical orthogonal function (EOF) of daily zonal-mean geopotential anomalies on the 500-hPa level for the region south of 20°S (Bracegirdle et al., 2020). The anomalies are calculated based on Gerber et al. (2010). First, a time series of 500-hPa zonally mean  $\overline{\Phi}(\lambda,t)$  is taken. where  $\lambda$  and t refer to latitude and time at daily intervals, respectively, and the bar indicates zonal average. Then, for each day, we subtract the global mean of 500-hPa geopotential from  $\overline{\Phi}(\lambda,t)$  at each latitude, and 230 the resulting data is linearly detrended. Lastly, a slowly varying climatology  $\widetilde{\Phi}(\lambda,t)$  is subtracted to remove the seasonal cycle and the low-frequency nonlinear trends associated with known external forcings such as the ozone hole formation/recovery and global warming signal. Such  $\widetilde{\Phi}(\lambda,t)$  field is derived in two steps following Gerber et al. (2010). To avoid overfitting high-frequency noise, a 60-day low-pass filter is first applied to the detrended  $\overline{\Phi}(\lambda,t)$  along the t axis to retain only seasonal-scale 235 variability. Specifically, we apply the discrete Fourier transform to the time series and filter out components with frequencies higher than 1/60 days<sup>-1</sup>. The resulting smoothed time series is then reindexed by calendar day (d) and year (y). For each calendar day (e.g., Jan 1st, Jan 2nd, etc.), a 30-year low-pass filter is subsequently applied along the y axis to extract long-term variations. If the data span 240 fewer than 30 years, the average across all available years for that calendar day is used, resulting in a fixed, repeating annual cycle.

The resultant anomalies  $\overline{\Phi'}(\lambda,t)$  reflect the internal/natural variability. We can then obtain SAM as the first EOF of  $\overline{\Phi'}(\lambda,t)$  over 20–90°S. For the computation of EOFs,  $\overline{\Phi'}(\lambda,t)$  is weighted by  $\sqrt{\cos(\lambda)}$  to account for the decreasing distance between meridians toward the pole. The resultant leading EOF  $\mathbf{e}(\lambda)$  represents the spatial patterns of SAM, and its corresponding principal component time series PC(t) is referred to "SAM index", expressed in normalized form with zero mean and unit variance (Fig. 1a-b).

To quantify the SAM persistence, the decorrelation time scale is computed based on the autocorrelation function of the SAM index following Simpson et al. (2013a):

$$ACF(d,l) = \frac{\sum_{y=1}^{N-1} PC(d,y) PC(d+l,y)}{\sqrt{\sum_{y=1}^{N-1} PC(d,y)^2 \sum_{y=1}^{N-1} PC(d+l,y)^2}}.$$
 (1)

Here, the daily time series PC(t) is reindexed as a function of calendar day d (e.g., Jan 1st to Dec 31st) and year y, and N denotes the total number of years. Equation (1) computes the autocorrelation of PC between a given day d and a lagged day d+l, averaged over all available years. The ACF(d, l) is then smoothed over a 181-day window along the d axis (to smoothen daily fluctuations) using a Gaussian filter with a full width at half maximum of 42 days (standard deviation of 8 days). Finally, for each d, an exponential curve is fitted to the smoothed ACF(l) up to a lag of 50 days using the least squares method, and the e-folding time scale ( $\tau$ ) is then derived at which the exponential fit decreases to  $e^{-1}$  (Fig. 1c).

To provide a measure of sampling uncertainty of  $\tau$ , we perform 1,000 times of bootstrap resampling, each time redrawing all yearly PC(d, y) with replacement to form a new sample as large as the original sample size (N). Repeating the above ACF calculation for all bootstrap samples leads us to 1,000 values of  $\tau$  for a given day (Fig. 1c), showing its possible range.

Note that the above EOF analysis is performed separately for all datasets to identify SAM as the leading mode within each simulation, allowing for potential differences in its spatial structure across models.

Figure 1. Example of the SAM decorrelation timescale and eddy feedback strength calculation based on ERA5: (a) The first EOF pattern based on 500-hPa geopotential; (b) The associated first PC1 time series (only a partial segment is shown here); (c) Autocorrelation function (ACF) of the SAM index (smoothed with a Gaussian filter) shown for a given day of the year (black dashed), and an exponential fit (yellow). The e-folding timescale is denoted as τ. The calculation of ACF is repeated 1,000 times for the bootstrap samples (gray). (d) Same as (a) but based on vertically averaged zonal wind. (e) Lagged regression of the budget terms in Eqn. (3) onto the SAM index. (f) Eddy feedback strength b for lags 7–14 days.

## 270 **3.2** Tropospheric westerly jet position

The westerly jet position is diagnosed following Menzel et al. (2019) and Barnes and Polvani (2015) using the output on the native model grid. We first identify the latitude ( $\lambda_{max}$ ) of the maximum monthly zonally averaged 850-hPa zonal wind between 75°S and 10°S. Then, we apply a quadratic fit to the zonally averaged zonal wind at  $\lambda_{max}$  and at the two adjacent latitudes to the north and south. The latitude corresponding to the maximum value of this quadratic fit defines the position of the tropospheric westerly jet.

#### 3.3 Contribution of atmospheric eddy feedback strength to SAM persistence

Various methods have been proposed to quantify the strength of tropospheric eddy-mean flow feedback. We adopt the approach of Simpson et al. (2013b), as it has been applied to CMIP5 model evaluation and 280 is highly correlated with the summertime SAM persistence bias (coefficient of 0.83). This approach estimates the contribution of eddy momentum flux convergence to the tendency of SAM-associated westerly wind anomalies. Therefore, within this framework, SAM is alternatively described by the first EOF of vertically averaged (pressure weighted) zonal-mean zonal wind anomalies, deseasonalized and detrended, over 20-90°S. The resultant EOF latitudinal pattern (e) and associated PC time series are defined such that the former has units of m s<sup>-1</sup> (Fig. 1d), the latter has unit variance, and their 285 multiplication reconstructs the SAM-associated zonal wind anomaly fields in latitude and time space. This shift from a definition of the SAM persistence timescale using geopotential height to the zonal wind for the estimation of the eddy-mean flow feedback is based on the standard assumption that geostrophic equilibrium provides a good approximation of the relevant variables. However, ageostrophic terms can also contribute to SAM persistence, introducing limitations to this hypothesis (Vishny et al. 2024; Smith 290 et al. 2024). For simplicity and consistency with Simpson et al. (2013b) in their CMIP5 assessment, only three pressure levels of 850, 500, 250 hPa are utilized for this analysis.

A quantity or a forcing term (denoted as X as an example) associated with the SAM is derived by projecting it onto the EOF pattern ( $\mathbf{e}$ ) with the operator:

$$[\bar{X}]_s = \frac{[\bar{X}]We}{\sqrt{e^TWe}},$$
(2)

where the overbars denote the zonal mean, brackets indicate the vertical average,  $[\overline{X}]$  is a vector form of  $[\overline{X}](\lambda,t)$ , where  $\lambda$  and t are latitude and time, and  $\mathbf{W}$  is a matrix with diagonal elements equal to the  $\cos(\lambda)$  weighting (Simpson et al. 2013b). The resultant  $[\overline{X}]_s$  is a time series. How strongly the eddy forcing sustains the SAM wind anomalies is then estimated by projecting the vertically and zonally averaged zonal momentum equation onto  $\mathbf{e}$ :

$$\frac{\partial [\bar{u}]_s}{\partial t} = [\bar{m}]_s + [\bar{F}]_s, \tag{3}$$

$$[\overline{m}]_s = -\left[\frac{1}{a\cos^2\lambda} \frac{\partial (\overline{u'v'}\cos^2\lambda)}{\partial \lambda}\right]_s,$$
 (4)

where  $[\bar{m}]_s$  is the eddy momentum flux convergence attributed to SAM, u' and v' are the deviation of the zonal and meridional velocities from their zonal means, respectively, and are calculated based on the instantaneous fields at 6-hourly intervals before being converted to daily means, a is the Earth radius, and  $[\bar{F}]_s$  represents all the residual momentum forcing associated with SAM. Note that Equation (3) assumes that the sum of individual projected forcing terms on the right-hand side is in balance with the tendency of the SAM anomalies. Simpson et al. (2013b) demonstrate the validity of this assumption.

Lorenz and Hartmann (2001) hypothesized that the eddy forcing of the SAM consists of a random component and a feedback component that depends linearly on the pre-existing state of SAM,  $[\overline{m}]_s =$ 310  $\widetilde{m} + b[\overline{u}]_s$ , where b denotes the eddy feedback strength. To obtain b, Simpson et al. (2013b) performed the lagged linear regressions of  $[\overline{m}]_s$  and  $[\overline{u}]_s$  onto the SAM index PC(t), such that for a lag day l,  $[\overline{m}]_s(t+l) \approx \beta_m(l)PC(t)$  and  $[\overline{u}]_s(t+l) \approx \beta_u(l)PC(t)$ , where  $\beta_m$  and  $\beta_u$  are the regression coefficients (Fig. 1e). Accordingly, the eddy forcing of SAM at lag l,  $[\overline{m}]_s(t+l)$ , can be expressed as  $\beta_m(l)PC(t) = \beta_{\widetilde{m}}(l)PC(t) + b\beta_u(l)PC(t)$ . Assuming that the random component of the eddy forcing is uncorrelated at sufficiently large positive lags, i.e.,  $\beta_{\widetilde{m}} \approx 0$ , we can estimate the eddy feedback strength as a function of lag days (l) by

$$b(l) = \frac{\beta_m(l)}{\beta_u(l)}.$$

Following Simpson et al. (2013b), b is averaged over lags 7 to 14 days (Fig. 1f). The approach followed here assumes that analyzing only the first PC is a good approximation to study SAM persistence. While 320 the PCs are uncorrelated on short timescales (by construction), this is not the case at longer lags and the coupling between the first two components influences SAM persistence (Sheshadri and Plumb 2017, Lubis and Hassanzadeh 2021, and Lubis and Hassanzadeh 2023). Analyzing only the first PC brings thus clear limitations in our analysis. Furthermore, positive regression coefficients could be caused by nonstationarity of the series and in particular by interaction with the stratosphere and not just by eddy mean flow interactions. This introduces biases in the estimate of eddy feedback, particularly in late spring and summer (Byrne et al. 2016, Byrne et al. 2017), although this does not necessarily prevent using the regression method (Ma et al. 2017). The methodology is thus imperfect, but it provides an interpretative framework for the difference between the simulations and allows a comparison with earlier studies.

#### 330 3.4 Contribution of surface friction to SAM persistence

While eddy momentum flux convergence primarily supports the persistence of SAM, surface friction predominantly acts to dissipate SAM anomalies. Since the friction forcing is not a standard output of EERIE simulations, we estimate it from the available variable: the turbulent wind stress in the eastward direction (in units of N m<sup>-2</sup>). By assuming the turbulent wind stress is zero at the model top, we can estimate the friction as  $\frac{0-\rho_0^{-1}WS_0}{H_0}$ , where  $WS_0$  indicates the daily-mean eastward turbulent stress near the surface, resulting from turbulent atmospheric eddies (due to the roughness of the surface) and turbulent orographic form drag. For simplicity, we assume fixed values of the air density  $\rho_0 = 1.204 \text{ kg/m}^3$  and the atmosphere column depth  $H_0=8,464$  meters here. Following a similar approach for calculating  $[\overline{m}]_s$ , we projected the result onto the EOF pattern (e) to obtain the frictional forcing for the SAM zonal wind anomalies, denoted as  $[\overline{f}]_s$ . To provide an alternative measure of friction forcing and verify the estimation, the residual term of Eq. (3),  $[\overline{F}]_s$ , is also computed based on the estimates of the acceleration and eddy momentum flux convergence, given the dominance of friction in this residual as shown in Simpson et al. (2013b).

It is important to note that the projection values of all budget terms are resolution (number of data points)dependent, as defined by Eq. (2). Therefore, their magnitudes are not directly comparable across datasets
with differing resolutions unless regridded to a common grid, as done here.

#### 4. Results

## 4.1 Model performance for SAM persistence

Figure 2 compares the performance of EERIE and CMIP6 models in representing SAM persistence, measured by the decorrelation time scale (τ). Consistent with Bracegirdle et al. (2020), CMIP6 models tend to overestimate SAM persistence compared to the reanalysis data analyzed over the same historical period (1980–2014). On the annual mean, CMIP6 presents a median value of 11 days, while the ERA5

shows a  $\tau$  of 8 days. A reduced bias is found for EERIE coupled simulations with a median  $\tau$  of 9 days, although the distribution spread is still large, suggesting a large inter-model variability. Among these simulations, positive biases persist in the IFS-FESOM2 1950control ( $\tau$ =13) and historical ( $\tau$ =11) runs, and ICON historical simulation show negative bias ( $\tau$ =6). Meanwhile, HadGEM3 piControl ( $\tau$ =9) and ICON 1950control ( $\tau$ =8) runs are closer to ERA5. Given that some of these simulations are run under a pre-industrial 1850s' or 1950s' forcing, we also examine the result based on an earlier-period ERA5 (1958–1978), for which τ increases to 10 days. Note, however, that there is relatively less confidence in the accuracy of the value of the SAM in ERA5 prior to the satellite era. Nevertheless, EERIE still show an improved agreement with ERA5 as their  $\tau$  fully cover the uncertainty ranges of ERA5 for both periods. During the austral early summer (NDJ), the overestimation of SAM persistence in CMIP6 is more pronounced with a longer tail of  $\tau$  distribution. The maximum and median  $\tau$  in CMIP6 is 32 days and 17 days, respectively, compared to the ERA5 value of 11 days for the same historical period. Compared to CMIP6, EERIE coupled simulations exhibit some improvement with the maximum and median values dropping to 20 days and 16 days, respectively. However, the spread among different EERIE simulations remains large; while the positively biased τ are mostly captured by IFS-FESOM2, ICON tends to exhibit much smaller  $\tau$  than ERA5 at 6-7 days.

Interestingly, the atmosphere-only EERIE simulations (IFS-AMIP) generally outperform the ocean-coupled runs, exhibiting a reduced positive bias in  $\tau$  compared to their coupled versions (IFS-FESOM2) and a much smaller spread. This suggests that the prescribed historical SST boundary condition serves a strong physical constraint on the SAM persistence. With all five members considered, the simulated  $\tau$  at 28 km is still positively biased for both annual and austral-summer means, but the biases do not exceed more than 4.5 days and at least one member presents almost identical values (8 days annually and 11 days in NDJ) to ERA5 (1980-2014). Refining the atmospheric resolution from 28 km to 9 km suggests a lowering of the SAM decorrelation timescale, with  $\tau$  of 8 days annually and 10 days in NDJ. However, the difference may not be robust, as the bootstrapped error bars of both resolutions overlap.

Figure 2. Distribution of τ (days) in CMIP6, EERIE coupled, and EERIE atmosphere-only (AMIP) simulations. CMIP6 and EERIE AMIP are both historical simulations, with a fixed period indicated in the x-axis labels, and the EERIE coupled simulations cover varied periods as indicated in Table 1. ERA5 is analyzed for two time periods. CMIP6 results from 31 experiments are presented in violin plot, in which the width indicates the density of the data points, the thin gray vertical box in the middle shows the 25th –75th quantiles, and the white dot presents the median. For the rest, error bars are added wherever applicable to show the ±1 standard deviation of τ from the 1,000 bootstrap resampling.

## 4.2 The relationship between jet location and $\tau$

The bias relationship between westerly jet location ( $\lambda_0$ ) and SAM decorrelation timescale ( $\tau$ ) is then revisited. Similar to their predecessors, CMIP6 models show a positive correlation between  $\lambda_0$  and  $\tau$ , that is, models with a more equatorward jet location tend to exhibit a more persistent SAM. Consistent with Simpson & Polvani's (2016) result based on CMIP5 models, the slope of the linear fit is larger during NDJ, indicating a larger variation in  $\tau$  given the same variation in  $\lambda_0$  during this season.

As EERIE results suggest that higher resolution may reduce persistence biases, we examine the model resolution of each CMIP6 simulation. However, there appears no strong or clear relationship between the model resolution and the model biases in either  $\tau$  or  $\lambda$  (the conclusion holds for both latitudinal and longitudinal resolutions and for both atmospheric and oceanic components, although only the atmospheric latitudinal resolution is expressed in Fig. 3). A potential dependency on resolution could be obscured in the CMIP6 ensemble by other compensating factors arising from different model configurations and system designs. However, it is also possible that resolution-driven improvements have plateaued within the typical grid size range of current GCMs (e.g., CMIP6). For instance, based on simplified atmospheric GCMs with idealized forcing, Gerber et al. (2008) found that the decorrelation timescale of the annular mode is unrealistically large at a coarse resolution of T21 (5.6°). While such a bias was notably reduced by refining the model resolution to T42 (2.8°), no further improvement was shown with a higher resolution of T85 (1.4°) and the  $\tau$  converges to a still positively biased value. No test was performed in this study to determine if  $\tau$  is improved again at even higher resolution or if the plateau continues.

On the annual mean, EERIE simulations generally fall within a region smaller than that covered by CMIP6, with the IFS-FESOM *1950control* being the worst performing experiment among the EERIE simulations (Fig. 3a), showing both the greatest positive bias in  $\tau$  and  $\lambda_0$ . For NDJ, the spread of EERIE clearly shifts toward a lower  $\tau$ , closer to ERA5's  $\tau$  compared to other CMIP6 exhibiting a similar jet location. In all, a positive  $\lambda_0$ - $\tau$  relationship remains and appears stronger in summertime across EERIE models (Fig. 3). The most skillful EERIE simulations for the SAM persistence, IFS-AMIP, all well capture the jet location (with a bias 

Figure 3. Scatter plot of climatological jet latitude ( $^{\circ}$ ) versus SAM decorrelation timescale  $\tau$  (days; error bar indicates  $\pm 1$  standard deviation from the bootstrapping) for (a) annual and (b) early-summer (NDJ) means in the Southern Hemisphere. Green crosses are based on CMIP6 historical simulations (colored by their latitudinal atmospheric resolution). Model names are not labeled here for visual clarity, but details are provided in Supplementary Table 1. ERA5 reanalysis and EERIE simulations are indicated as in the legend. Vertical and horizontal black lines are the ERA5 values. The green dotted straight line is the linear least-squares regression fit for CMIP6 models (slope is denoted as m, and Pearson correlation coefficient r is expressed in bold if statistically significant with the p value <0.05 in green in the top left corner). Similarly, the black dotted fitted line is for all EERIE simulations.

## 4.3 Sensitivities to varying SST boundary conditions

EERIE simulations demonstrate a reduced bias in summertime SAM persistence compared to CMIP6, but identifying the cause is challenging due to variability in model systems. Although CMIP6 results show no clear link between model resolution and performance in  $\tau$  and  $\lambda_0$ , the higher resolution in EERIE remains one possible contributing factor to such an improvement. One piece of evidence is the reduction in  $\tau$  when transitioning from a 28-km to a 9-km model grid size using a consistent IFS model. Another

possibility is that the new generation of models in EERIE improves model physics, reducing the biases in processes that resulted in a overly-persistent SAM in earlier CMIP-like GCMs. In addition, EERIE begins to explicitly resolve the ocean mesoscales, which are parameterized in CMIP6, though the resulting impacts on SAM persistence have not been investigated. To explore these possibilities within a controlled framework, this section focuses on EERIE atmosphere-only sensitivity experiments with and without SST eddies (ObsSST vs. NoEddies) at two model resolutions.

We first focus on the 28-km simulations. Regarding the seasonal variation of τ (Fig. 4a), the NoEddies experiments exhibit intermingled patterns overlapping with those of ObsSST. Although their ensemble means suggest a slight reduction in τ (by approximately 2 days) in NDJ in the absence of ocean eddies—hinting that mesoscale SST features may help sustain SAM persistence—this difference is not statistically significant at 95% confidence level. For the 9-km configuration, the subtle impact of ocean eddies is not observed as NoEddies shows no clear changes in τ from ObsSST, and both show smaller τ than the 28-km counterparts (Fig. 4b).

- All these sensitivity experiments show a slightly poleward biased jet latitude compared to ERA5 (within 1°) during NDJ, and ObsSST are generally less biased than NoEddies (Fig. 4c). While this seems to be in agreement with the literature that a more southward-shifted jet is associated with a longer SAM persistence, the correlation between  $\lambda_0$  and  $\tau$  is weak (with a correlation coefficient of 0.03) across all simulations in the IFS-AMIP configurations.
- Compared to  $\lambda_0$ , the metric eddy feedback strength b shows a much stronger correlation with SAM persistence  $\tau$ , with a higher correlation coefficient of 0.52 and a lower p-value of 0.08 (Fig. 4d), suggesting it may be a more informative indicator of SAM persistence in this configuration. Meanwhile, the surface friction and  $\tau$  exhibit a negative correlation (Fig. 4e) with a moderate correlation coefficient of -0.48 and p-value of 0.11. It is worth noting that our results using  $[\bar{f}]_s$  based on surface wind stress show qualitatively consistent patterns with those using the residual estimates,  $[\bar{F}]_s$ , across simulations despite some differences in the absolute values (Fig. S1a, b). A closer examination shows that the member with the largest value in  $[\bar{f}]_s$  is accompanied by the weakest eddy feedback b (red cross markers in Fig. 4d, e) and vice versa (red square). The opposite shifts of these two dominant mechanisms indicates an offset between each other, leading to subtle combined effects on the SAM persistence.

- Lorenz and Hartmann (2001) proposed that the eddy feedback can interact with the frictional impact to lengthen the effective timescale of SAM by  $\frac{t_f}{(1-bt_f)}$ , where  $t_f$  is the damping timescale. Here, we estimate  $t_f$  by taking the ratio between the regressed  $[\bar{u}]_s$  (in unit of m/s) and the regressed  $[\bar{F}]_s$  (unit of m/s<sup>2</sup>) averaged over the 7-14 lag days, which gives a value of 8.6 days for ERA5 (close to the 8.9 days in Lorenz and Hartmann (2001)). This metric correlates with  $\tau$  more strongly than b or  $[\bar{f}]_s([\bar{F}]_s)$  alone, showing a
- higher correlation coefficient of 0.61 and a lower p-value of 0.03 (Fig. S1d). This result points to the importance to assess the joint/net impact of the competing dominant mechanisms.
- However, although those metrics explain some of the differences between individual experiments, none of them shows systematic differences between ObsSST and NoEddies and none clearly accounts for the significant reduction in τ when the model grid size is refined from 28 km to 9 km. Considering the large variability in the 28-km ensemble members, one member at 9 km may be not enough to identify the influence of the resolution. Additional simulations and different experimental approaches may be required to confirm the underlying cause for the observed model grid spacing dependency.

Figure 4. Analysis of the IFS-AMIP idealized experiments (black for ObsSST and red for NoEddies; yellow for ERA5 as reference):

(a) SAM decorrelation timescale (τ) as a function of month for 28km simulations (dashed for individual ensemble members and solid for ensemble means). (b) Similar to (a) but for 9 km experiments (shades for the ±1 standard deviation of τ from the 1,000 bootstrap resampling). (c) Scatter plot of τ (days; y-axis) and westerly jet latitude (x-axis; filled-color markers for 28 km; hollow stars for 9 km simulations). (d)–(e) Similar to (c) but with x-axis variable replaced with the eddy feedback strength and frictional impact, respectively. In (b)–(d), the same marker shape indicates the same ensemble member. The gray dotted line represents the linear regression fit, and the correlation coefficient and p-value are indicated in the top-right corner.

#### 5 Discussion and conclusions

This study assesses the performance of new high-resolution global model simulations developed under the EERIE project in capturing the persistence of the Southern Annular Mode (SAM), a leading mode of climate variability in the Southern Hemisphere. EERIE simulations are conducted with a model grid size of 9–28 km for the atmosphere and 5–13 km for the ocean. The persistence of the SAM is assessed using the decorrelation timescale of the SAM index  $(\tau)$ , for which CMIP GCMs have historically exhibited a

systematic positive bias (overly persistent) in austral summer, often correlated with a climatological westerly jet that is too equatorward. Our conclusions and discussion based on the phase 1 preliminary simulations of the EERIE models are organized into two subsections: (1) the performance of coupled simulations, and (2) the performance of atmosphere-only (AMIP) simulations and insights obtained from the sensitivity experiments with varied SST boundary conditions under the AMIP setup.

#### **5.1 EERIE Coupled simulations**

Compared to CMIP6, the EERIE coupled simulations show improvement in representing the SAM 500 persistence. Although the inter-model variability remains large, the annual τ distribution of EERIE coupled simulations clearly shifts to lower biases with a median value of 9 days, closer to the ERA5 value of 8 days than the CMIP6's median of 11 days. During early summer, the pronounced long tail of  $\tau$  in CMIP6 simulations is also noticeably reduced in EERIEs, with the former ranging from 9 to 32 days (median: 16 days) and the latter ranging from 9 to 17 days (median: 14 days) closer to ERA5's 11 days. The relationship between biases in the westerly jet location ( $\lambda_0$ ) and  $\tau$  remains positively correlated in 505 EERIE simulations as has been documented for CMIP-like models. Consistently, the smaller bias for  $\tau$  in EERIE simulations is accompanied by improved representation of  $\lambda_0$  compared to CMIP6. However, some CMIP6 models capture jet locations similar to EERIE, yet still perform worse for  $\tau$ , suggesting other factors are at play. While the improvement of EERIE models compared to CMIP6 indicates that 510 increased resolution can offer benefits, the varied skills within CMIP6 in representing either  $\lambda_0$  or  $\tau$  do not show a clear dependency on the model resolution. It is possible that the impact of resolution is outweighed by other varying factors in CMIP6, or that resolution-driven benefits have plateaued within the grid-size ranges in current CMIP6 and require more substantial resolution refinement to emerge.

## 5.2 EERIE Atmosphere-only simulations

Among EERIE simulations, the IFS-AMIP runs with prescribed historical SST and sea ice boundary conditions show the optimal performance in both SAM persistence and westerly jet location, with smaller spreads and closer values to ERA5 than the coupled runs. This highlights the importance of accurately representing sea surface thermal conditions to improve the simulation of these large-scale atmospheric

quantities. While Sen Gupta and England (2006) showed that air-sea coupling is critical for modulating the SAM—albeit focusing on interseasonal timescales, which are longer than the intraseasonal scale investigated here—our results suggest that atmosphere-ocean coupling plays a secondary role. Instead, SST biases introduced by the coupling—an ongoing challenge in coupled GCMs (Zhang et al., 2023)—appear to be more influential.

For the AMIP historical simulations, the λ<sub>0</sub>-τ bias relationship is virtually absent. We speculate that when the jet is already well captured (all AMIP runs are with <1° bias) and SSTs are prescribed, other second-order processes may come into play to affect τ. Indeed, we find that the metrics of atmospheric eddymean feedback strength, surface friction and their joint effect correlate more strongly with τ than with λ<sub>0</sub> in the AMIP configurations, highlighting the importance of these mechanisms on SAM persistence. However, these metrics cannot fully explain the clear reduction of τ when the model resolution is refined from 28 km to 9 km using the same atmospheric model.

Finally, the thermodynamic impact from the ocean mesoscale features is explored via idealized AMIP experiments by filtering out the transient ocean eddies (NoEddies) in the SST boundary conditions. While the difference between the 28-km ensemble means of ObsSST and NoEddies imply that the ocean mesoscale SST features may help to maintain the SAM anomalies (increase  $\tau$  by roughly 2 days) in early summer, such an impact is not statistically significant and is not captured in the 9-km simulations. Among the 28-km members, we also do not see a systematic change of eddy feedback or surface friction due to the presence or absence of ocean eddies in the SST field. The critical role of oceanic mesoscale eddies in the Southern Ocean climate system is well documented. While their local impact on the atmospheric boundary layer is well established, their direct influence in modulating large-scale modes such as the SAM appears limited under our AMIP setup without air-sea coupling. A similar conclusion was obtained by Purich et al. (2021) with a coarser coupled GCM (model resolution of ~130 km), ACCESS1.0. They found that suppressing Southern Ocean SST variability by restoring the SST to the monthly mean patterns does not impact SAM persistence in their simulations, but they also concluded that eddy-resolving models are required to properly capture the air-sea feedbacks in the Southern Hemisphere.

The superior performance of the AMIP compared to coupled simulations might suggest that model skill in representing SAM persistence gains little from two-way ocean—atmosphere coupling or explicit

resolving ocean mesoscale features. Our hypothesis is that while coupled models offer a more physically consistent representation of the climate system, they also tend to introduce SST biases—potentially due to under-tuning in high-resolution configurations or imbalances in the coupling process. In fact, previous studies have shown that eddy-permitting models can exhibit larger SST biases than either coarser models with parameterized eddy fluxes or fully eddy-rich models (e.g., Storkey et al. 2025). Reducing SST biases remains essential for advancing the representation of SAM and Southern Hemisphere climate variability. The large variability among ensemble members highlights the intricate mechanisms behind SAM persistence in GCMs. It urges deeper investigation and alternative approaches to resolve outstanding questions regarding the atmospheric variability in the Southern Hemisphere. For example, this study only considers the zonally averaged properties, but non-zonal components likely play important roles in shaping SAM characteristics and hence their representation in GCMs (e.g., Barnes and Hartmann, 2010; Sen Gupta and England, 2006). Nevertheless, the general improvements seen in the phase 1 simulations of the EERIE coupled models present promising results in addressing the long-standing GCM biases in SAM persistence, especially considering the challenges in optimally configuring high-resolution models (i.e., tuning) and the lack of community experience in doing so. Furthermore, the controlled framework of the IFS-AMIP idealized eddy-rich experiments offers significant potential for enhancing our understanding of atmospheric responses to ocean mesoscales.

## 565 Author Contributions

TC performed the majority of the data analysis and wrote the initial draft of the manuscript. HG led the conceptual design of the study, guided the interpretation of the results, and oversaw the project. CD contributed to specific aspects of the data analysis. TC, HG, CD, and SL collaboratively addressed reviewers' comments and refined the manuscript. CR, MA, MR, RG, and JS contributed to model development, conducted simulations, and prepared the data. KS provided essential support in handling the CMIP6 data. All co-authors contributed throughout the project, engaged in scientific discussions, and approved the final version of the manuscript for submission.

#### Acknowledgments

This publication is part of the EERIE project funded by the European Union. Views and opinions expressed are however those of the author(s) only and do not necessarily reflect those of the European Union or the European Climate Infrastructure and Environment Executive Agency (CINEA). Neither the European Union nor the granting authority can be held responsible for them. This work has received funding from the Swiss State Secretariat for Education, Research and Innovation (SERI) under contract #22.00366. This work was funded by UK Research and Innovation (UKRI) under the UK government's Horizon Europe funding guarantee (grant number 10057890, 10049639, 10040510, 10040984). Hugues Goosse is Research Director with the Fonds de la Recherche Scientifique (F.R.S. -FNRS). The authors also thank Francesco Ragone, Isla Simpson, Matthew Patterson, Thomas Bracegirdle for providing constructive suggestions or clarification about the methodologies employed in their works. We gratefully acknowledge the two anonymous reviewers for their constructive and valuable feedback.

#### 585 Data Availability Statement

All EERIE simulation outputs are publicly accessible at <a href="https://eerie.cloud.dkrz.de">https://eerie.cloud.dkrz.de</a> and Wachsmann et al. (2024). The calculation of EOFs was performed using the publicly available Python package by Dawson (2016). All scripts used for the analysis and figure generation will be shared in the GitHub repository under the EERIE project (<a href="https://github.com/eerie-project">https://github.com/eerie-project</a>) upon acceptance of the publication.

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
