# Peer review of "Southern Annular Mode Persistence and Westerly Jet: A Reassessment Using High-Resolution Global Models"

_EGUsphere, 2025_

## Referee Comment (RC1)

**Review of "Southern Annular Mode Persistence and Westerly Jet: A Reassessment Using High-Resolution Global Models" by Ting-Chen Chen et al.**

**Overall Assessment**

This study explores the atmospheric and oceanic influences on modelled SAM persistence and its relationship with the latitude of the mid-latitude jet. The authors note the longstanding issue that CMIP models (including the latest suite: CMIP6) overestimate SAM persistence (quantified using the decorrelation timescale), particularly in early austral summer, which appears to be much improved when using high-resolution, eddy-resolving simulations from the EERIE project. This appears to be in part due to more realistic simulation of the jet position/distribution (CMIP models have tended to be too equatorward biased) but the importance of accurate SST representation is also clear. In fact, the authors show that AMIP model experiments of the EERIE simulations perform better than coupled experiments in terms of more realistically representing the jet position and SAM decorrelation timescale.

Enhanced resolution of the EERIE simulations likely plays a role in the improvement relative to CMIP6 models but also improved model physics, particularly concerning ocean mesoscale eddies which appear to slightly enhance the SAM decorrelation timescale in early summer (at least for simulations run at 28 km resolution) according to sensitivity simulations performed. However, cancellation effects (e.g., atmospheric eddy feedback strength versus surface friction) make it difficult to ascertain which aspects help improve modelling of the SAM persistence. For instance, the decorrelation timescale is more realistic still in 9 versus 28 km AMIP simulations, yet the role of ocean mesoscale eddies in enhancing SAM persistence is not evident at this finer resolution.

I found the study to be very well written, organised and logically structured. The Figures are clearly presented and straightforward to understand and the step-by-step computation of the different diagnostics examined will I think be much appreciated by many readers. The conclusions drawn are supported by the results shown, so I can recommend this be accepted for publication in *Weather and Climate Dynamics*. I include just a few comments for the authors to consider prior to acceptance.

**General Comments**

I only had one main consideration for the authors which I found lacking in the paper. That is information of which CMIP6 models were considered (if supplementary information were to be provided, a table for this would be warranted). That is not to say that knowledge of which models lie where in the distributions shown (Figures 2 and 3) are important in understanding this work. But for others reading, it might be useful for them to know and the best way I feel to include this would be to show similar Figures with individual CMIP6 models indicated in supplementary information.

Nevertheless, knowledge of which CMIP6 models lie where could help others or even the authors to comment upon whether commonalities such as shared model components or known biases more widely within the climate system might influence the results. So, I would encourage the authors to think about providing this information.

**Specific Comments**

**L15:** "a critical driver" → "a leading mode"? I wouldn't consider the SAM to a driver itself, but more of a reflection of driving influences. Expressing it as a leading mode would be more scientifically accurate and consistent with earlier literature (e.g., Marshall, 2003; Marshall et al., 2022).

**L340-342:** How reliable is the SAM derived from ERA5 pre-satellite era when comparing with the EERIE coupled simulations? Presumably the SAM is more reliably reconstructed after 1979 from ERA5 but maybe difficult to quantify how much of an improvement there would be. It may be worthy of further comment or caveating, however?

**L343-344:** Suggesting that a relative minority of models are considerably worse in representing SAM decorrelation timescale than the rest of the pack. Did the authors investigate why this might be or were they at least able to note some commonalities in the most unrealistic models that might point to the cause(s)? For instance, could there be an association between too equatorward jet position and shared model components? Or factors that may give plausibly give rise to the issue of realistic eddy feedback strength? It may be beyond the scope of the paper to delve into this, but others reading might be encouraged to look into this.

**L477:** "…a key driver…" → Again I think '…a leading mode…" would be more technically correct.

**Technical Corrections**

**L52-53:** "spring (MAM) and summer (JJA)". → "autumn (MAM) and winter (JJA)".

**L153:** "observation" → "observations"

**Table 1:** Some font size inconsistencies noted.

**L245:** Tabulation before "Finally…"

**Additional References**

Marshall, G. J. (2003). Trends in the Southern Annular Mode from observations and reanalyses. *Journal of climate*, *16*(24), 4134-4143, https://doi.org/10.1175/1520-0442(2003)016<4134:TITSAM>2.0.CO;2.

Marshall, G. J., Fogt, R. L., Turner, J., & Clem, K. R. (2022). Can current reanalyses accurately portray changes in Southern Annular Mode structure prior to 1979?. *Climate Dynamics*, *59*(11), 3717-3740, https://doi.org/10.1007/s00382-022-06292-3.

---

## Author Comment (AC1)

**Response** to referees' comments on "Southern Annular Mode Persistence and Westerly Jet: A Reassessment Using High-Resolution Global Models" by Chen et al.
MS No.: egusphere-2025-666
MS type: Research article

**Referee 1 (RC1)**

**Overall Assessment**

This study explores the atmospheric and oceanic influences on modelled SAM persistence and its relationship with the latitude of the mid-latitude jet. The authors note the longstanding issue that CMIP models (including the latest suite: CMIP6) overestimate SAM persistence (quantified using the decorrelation timescale), particularly in early austral summer, which appears to be much improved when using high-resolution, eddy-resolving simulations from the EERIE project. This appears to be in part due to more realistic simulation of the jet position/distribution (CMIP models have tended to be too equatorward biased) but the importance of accurate SST representation is also clear. In fact, the authors show that AMIP model experiments of the EERIE simulations perform better than coupled experiments in terms of more realistically representing the jet position and SAM decorrelation timescale. Enhanced resolution of the EERIE simulations likely plays a role in the improvement relative to CMIP6 models but also improved model physics, particularly concerning ocean mesoscale eddies which appear to slightly enhance the SAM decorrelation timescale in early summer (at least for simulations run at 28 km resolution) according to sensitivity simulations performed. However, cancellation effects (e.g., atmospheric eddy feedback strength versus surface friction) make it difficult to ascertain which aspects help improve modelling of the SAM persistence. For instance, the decorrelation timescale is more realistic still in 9 versus 28 km AMIP simulations, yet the role of ocean mesoscale eddies in enhancing SAM persistence is not evident at this finer resolution. I found the study to be very well written, organised and logically structured. The Figures are clearly presented and straightforward to understand and the step-by-step computation of the different diagnostics examined will I think be much appreciated by many readers. The conclusions drawn are supported by the results shown, so I can recommend this be accepted for publication in Weather and Climate Dynamics. I include just a few comments for the authors to consider prior to acceptance.

**General Comments**

I only had one main consideration for the authors which I found lacking in the paper. That is information of which CMIP6 models were considered (if supplementary information were to be provided, a table for this would be warranted). That is not to say that knowledge of which models lie where in the distributions shown (Figures 2 and 3) are important in understanding this work. But for others reading, it might be useful for them to know and the best way I feel to include this would be to show similar Figures with individual CMIP6 models indicated in supplementary information. Nevertheless, knowledge of which CMIP6 models lie where could help others or even the authors to comment upon whether commonalities such as shared model components or known biases more widely within the climate system might influence the results. So, I would encourage the authors to think about providing this information.

We would like to thank the reviewer for their insightful comments and will take them into account for the revised version of the manuscript.

Regarding the comment on the CMIP6 table, the following table will be added in the supplement:

| Model | τ Annual | τ NDJ | Jet latitude Annual | Jet latitude NDJ |
|---|---|---|---|---|
| TaiESM1 | 8.2 | 11.6 | -52.4 | -50.4 |
| AWI-ESM-1-1-LR | 12.0 | 21.2 | -49.0 | -50.4 |
| AWI-ESM-1-REcoM | 14.1 | 25.1 | -49.7 | -50.4 |
| BCC-CSM2-MR | 11.2 | 21.5 | -51.1 | -50.3 |
| BCC-ESM1 | 14.2 | 30.2 | -50.5 | -49.5 |
| FGOALS-f3-L | 11.4 | 16.6 | -49.7 | -48.9 |
| FGOALS-g3 | 9.4 | 12.2 | -49.7 | -49.5 |
| CanESM5 | 10.2 | 14.5 | -49.6 | -49.8 |
| IITM-ESM | 13.2 | 22.5 | -47.0 | -47.0 |
| CNRM-CM6-1 | 11.3 | 18.7 | -48.2 | -48.4 |

| | | | |
|---|---|---|---|
| CNRM-CM6-1-HR | 9.8 | 12.6 | -45.8 | -46.2 |
| CNRM-ESM2-1 | 11.9 | 20.2 | -48.5 | -48.4 |
| ACCESS-CM2 | 13.9 | 27.0 | -50.1 | -49.6 |
| EC-Earth3 | 9.4 | 12.9 | -50.3 | -49.5 |
| MPI-ESM-1-2-HAM | 12.7 | 24.3 | -49.9 | -49.7 |
| INM-CM4-8 | 9.2 | 9.5 | -51.7 | -53.4 |
| INM-CM5-0 | 10.7 | 15.2 | -50.3 | -52.0 |
| IPSL-CM6A-LR | 12.7 | 22.9 | -49.2 | -48.9 |
| MIROC6 | 11.7 | 20.3 | -48.7 | -49.6 |
| MPI-ESM1-2-HR | 11.9 | 15.6 | -48.8 | -48.4 |
| MPI-ESM1-2-LR | 10.0 | 14.3 | -47.9 | -48.4 |
| MRI-ESM2-0 | 16.0 | 32.1 | -48.0 | -47.0 |
| GISS-E2-1-G (1) | 13.2 | 23.4 | -51.1 | -50.9 |
| GISS-E2-1-G (2) | 10.8 | 17.0 | -50.8 | -50.5 |
| CESM2 | 9.0 | 13.4 | -51.9 | -52.3 |
| CESM2-FV2 | 9.3 | 13.9 | -52.7 | -52.2 |
| CESM2-WACCM | 11.6 | 16.3 | -52.0 | -51.9 |
| CESM2-WACCM-FV2 | 11.18 | 17.31 | -52.25 | -51.70 |
| NorESM2-LM | 7.09 | 9.60 | -52.64 | -51.60 |
| NorESM2-MM | 9.09 | 13.03 | -52.56 | -52.16 |
| GFDL-CM4 | 9.24 | 12.43 | -48.98 | -48.68 |
| ERA5 | 7.88 | 10.92 | -51.11 | -50.39 |

Table S1: Annual and summer (DJF) decorrelation timescale and westerly jet position for the 31 studied CMIP6 models

**Specific Comments**

L15: "a critical driver" → "a leading mode"? I wouldn't consider the SAM to a driver itself, but more of a reflection of driving influences. Expressing it as a leading mode would be more scientifically accurate and consistent with earlier literature (e.g., Marshall, 2003; Marshall et al., 2022).

We thank the reviewer and agree with this comment. We will change the sentence as suggested.

L340-342: How reliable is the SAM derived from ERA5 pre-satellite era when comparing with the EERIE coupled simulations? Presumably the SAM is more reliably reconstructed after 1979 from ERA5 but maybe difficult to quantify how much of an improvement there would be. It may be worthy of further comment or caveating, however?

We agree that ERA5 is less reliable in the pre-satellite era and will comment on this accordingly in the revised version. Nevertheless, compared to other reanalysis datasets that extend backwards in time beyond 1979, ERA5 has been shown to agree best with available station observations prior to 1979 and after (Marshall et al., 2022). We will add a comment on that in the method section:

L203: "Among the reanalysis products that extend backwards in time beyond 1979 (ERA5, 20CRv3, JRA-55), *ERA5 is found to agree best with station observations and produces good representation of SAM, both before and after the advent of satellite sounder data (Marshall et al., 2022)*"

We will also highlight that the SAM is more reliably reconstructed after 1979:

L343:" Note however that there is relatively less confidence in the accuracy of the value of the SAM in ERA5 prior to the satellite era."

L343-344: Suggesting that a relative minority of models are considerably worse in representing SAM decorrelation timescale than the rest of the pack. Did the authors investigate why this might be or were they at least able to note some commonalities in the most unrealistic models that might point to the cause(s)? For instance, could there be an association between too equatorward jet position and shared model components? Or

factors that may give plausibly give rise to the issue of realistic eddy feedback strength? It may be beyond the scope of the paper to delve into this, but others reading might be encouraged to look into this.

It is indeed something interesting to study. However, we did not identify obvious commonalities shared by the worse performing models (e.g. spatial resolution or warm bias over Antarctica or the Southern Ocean or shared components).

Previous studies using CMIP models have discussed the association between too equatorward jet position and models' issues to represent the SAM persistence (e.g., Bracegirdle 2020, Zhang 2021) and the related role of the eddy feedback strength. However, we have the feeling that a comprehensive study of the shared components of these models falls beyond the scope of the study, which focuses on the added value of the high-resolution EERIE models.

L477: "...a key driver..." → Again I think '...a leading mode...' would be more technically correct.

This will be replaced in the revised version:

L 476-478: *This study assesses the performance of new high-resolution global model simulations developed under the EERIE project in capturing the persistence of the Southern Annular Mode (SAM), **a leading mode** of climate variability in the Southern Hemisphere.*

**Technical Corrections**

L52-53: "spring (MAM) and summer (JJA)". → "autumn (MAM) and winter (JJA)".

L153: "observation" → "observations" Table 1: Some font size inconsistencies noted.

L245: Tabulation before "Finally..."

The suggested technical corrections will be implemented in the revised version.

**Additional References**

Bracegirdle, T. J., Holmes, C. R., Hosking, J. S., Marshall, G. J., Osman, M., Patterson, M., & Rackow, T. (2020). Improvements in circumpolar Southern Hemisphere extratropical atmospheric circulation in CMIP6 compared to CMIP5. *Earth and Space Science*, *7*(6), e2019EA001065.

Marshall, G. J. (2003). Trends in the Southern Annular Mode from observations and reanalyses. *Journal of climate*, 16(24), 4134-4143, https://doi.org/10.1175/15200442(2003)016<4134:TITSAM>2.0.CO;2.

Marshall, G. J., Fogt, R. L., Turner, J., & Clem, K. R. (2022). Can current reanalyses accurately portray changes in Southern Annular Mode structure prior to 1979?. *Climate Dynamics*, 59(11), 3717-3740, https://doi.org/10.1007/s00382-022-06292-3.

Zhang, X., He, B., Liu, Y., Bao, Q., Zheng, F., Li, J., ... & Wu, G. (2022). Evaluation of the seasonality and spatial aspects of the Southern Annular Mode in CMIP6 models. *International Journal of Climatology*, *42*(7), 3820-3837.

**Referee 2 (RC2)**

"Southern Annular Mode Persistence and the Westerly Jet: A Reassessment Using High-Resolution Models" examines the relationship between SAM persistence, the climatological jet latitude, and the classic eddy-feedback parameter in both CMIP6 models and a new suite of high-ocean-resolution models (EERIE), with some added AMIP-style simulations with one EERIE model to help interpret the effects of increased resolution. The work finds that EERIE simulations have much lower bias in the SAM timescale than CMIP6, particularly in summer (traditionally the worst season). The bias is even lower in the AMIP simulations forced by observational SSTs, which suggests that ocean-atmospheric coupling may contribute to the bias. While these EERIE simulations have a lower bias and lower resolution than most CMIP6 models, the CMIP models do not show much dependence on horizontal resolution. Instead, previously established relationships relating the SAM timescale to the jet latitude seem to hold for the CMIP models. For EERIE models, this relationship breaks down, and the eddy-feedback parameter has better correlations with the annular mode timescale. When SST gradients are reduced in the AMIP style simulations, the persistence is reduced, although the cause is unclear.

I cannot recommend the paper to be published in its current form. With substantial revision and extended analysis, it could eventually be published, but the current state of the paper presents only a very marginal advancement in knowledge in the area of SAM timescales, and the results are challenging to interpret without more context in the literature and clearer interpretive frameworks.

Despite these criticisms, the paper does a few things well. First, I think the question is well-defined: what are the impacts of high-resolution atmosphere and ocean models, and their coupling, on SAM persistence? I also think they have the data available to address this question, but it needs to be much better utilized. They outline their methodology in a very reproducible way, and generally they follow the previous literature (to a point). The writing is of good quality and reasonably easy to follow.

My major concerns are summarized below; a detailed discussion follows. The novel contributions of this work are the analysis of high-resolution simulations, the SST sensitivity experiments, and the consideration of friction to explain intermodel differences. All of these contributions require serious improvement.

We would like to thank the Reviewer for the detailed and constructive evaluation of our manuscript and all the suggestions that we will include in the revised version. This additional material will broaden the scope of our study, which was initially focused on a mainly descriptive, standard evaluation of the SAM in a new set of high-resolution simulations. The

Reviewer underlines three novel contributions in our work and that all three require serious improvement. The way we will introduce those improvements is explained in detail below following each specific comment. The main changes are briefly summed up below:

1. We will discard spin-up simulations and replace the old shorter simulations with longer runs (ranging from 22-year to 65-year long) in our new analysis; and their SAM e-folding timescale estimation will be supplemented with the bootstrapping method.

2. We will expand the literature review and strengthen its connection to our results. This will include a broader discussion of the processes contributing to SAM persistence and an extended examination of the potential role of ocean mesoscales, providing a clearer justification for our AMIP sensitivity experiments using varied SST boundary conditions.

3. We will follow the reviewer's suggestions to modify our methods to estimate surface friction for SAM persistence and expand the relevant discussion.

4.We will adopt the suggested new method for the jet location estimation.

We also acknowledge that we cannot answer all the questions in a single study, but when it is not possible to have robust conclusions, we highlight it as well as the limitations of our study.

Regarding the analysis of high-resolution simulations: the simulations are all short (10 years) and frequently non-stationary (spin-up) or non-overlapping with the observational record. Given the long timescales required for SAM timescale convergence, the significant impacts of non-stationarity on the estimation of the timescale, and the potential for decadal and supra-decadal variability in the feedback itself (following the jet latitude), interpreting the difference between the EERIE simulations, ERA5, and CMIP6 is very challenging. Clearly the bias is reduced, but it is not clear at present whether this is due to artefacts, random chance, or physically meaningful reductions. This problem could be partially alleviated by carrying out the bootstrapping techniques used for the reanalysis for the EERIE simulations. Longer runs/overlapping time periods would be preferable, but given the computational expense involved the current simulations might be acceptable given appropriate explanation of the caveats involved.

Since the submission of our manuscript, new simulations are available. This allows us to analyze longer runs and to discard all the spin-up simulations that may have strong issues with stationarity. The changes of the simulations are highlighted in the table below:

| Institution | Alfred Wegener Institute (AWI) | Max Planck Institute (MPI-M) | Met Office (MO) | ECMWF |
|---|---|---|---|---|
| | Coupled atmosphere-ocean models (eddy-rich) | | | Atmospheric model |
| System name | IFS-FESOM2 | ICON | HadGEM3-GC5-EERIE | IFS |
| Simulations analyzed in the initial submission (segments period/length) | 1950spinup (31 yrs) 1950control (20 yrs) Historical (1950–1969) | 1950spinup Cycle 2 (11 yrs) | piControl (30 yrs) | Historical (1980–2023) |
| **Modified simulations in this revised manuscript** | **1950control (65 yrs) Historical (1950–2014)** | **1950control (22 yrs) Historical (1950–1971)** | **piControl (30 yrs)** | **Historical (1980–2023)** |

We also applied bootstrapping to the EERIE simulations wherever applicable. Among all simulations analyzed, only the ICON 1950control simulation failed to produce a convergent fit across the 1,000 bootstrapped resamples, likely due to variability in the underlying autocorrelation structure. Consequently, we will add the standard deviation of the e-folding timescale (τ) to Figure 2 in the revised manuscript for all simulations except ICON 1950control (as shown in Fig. R2-1 below).

[Figure]

Figure R2-1. Distribution of τ (days) in CMIP6, EERIE coupled, and EERIE atmosphere-only (AMIP) simulations. CMIP6 and EERIE AMIP are both historical simulations, with a fixed period indicated in the x-axis labels, and the EERIE coupled simulations cover varied periods as indicated in Table 1. ERA5 is analyzed for two time periods for references. CMIP6 results from 31 experiments are presented in violin plot, in which the width indicates the density of the data points, the thin gray vertical box in the middle shows the 25th–75th quantiles, and the white dot presents the median. For the rest, error bars are added wherever applicable to show the ±1 standard deviation of τ from the 1,000 bootstrap resamples.

Regarding the SST sensitivity experiments: these simulations require much clearer justification. At the moment, there is very little literature which would suggest that mesoscale SST gradients should affect the SAM persistence. There are some possible physical arguments, but they are not given here. One might argue that the fact that they do appear to influence the timescale in some small ensembles using one model is justification enough, but without a solid hypothesis to test, there is no definitive answer about why the change in timescale appears. It is entirely possible that it is by chance (no estimate of sampling uncertainty is provided). An incomplete argument discusses the role of surface friction, but it requires more explicit discussion of possible mechanisms. Some additional analysis on why persistence changes grounded in possible physical mechanisms would drastically strengthen the paper. It also needs to be much clearer why two different types of mesoscale features are included. My understanding of the feedback literature is that there is no reason to expect different results from the two, and they provide basically identical results.

As suggested by the Reviewer in a point below, we will add a paragraph in the revised version to better contextualize our work in the existing literature, specifically regarding the processes that control SAM persistence. After this paragraph, we will develop the justification of the sensitivity experiments (see below (*) for a suggestion of the text addressing those points).

Regarding the consideration of friction: The literature has established methods for estimating surface friction using the model output available to the authors, but they are not followed here. Instead, the authors estimate the frictional contributions in a way which is difficult to connect to established theory of SAM persistence and the feedback parameter they calculate, and in a way that also cannot be interpreted easily across model resolutions. Its physical units are not transformed to be consistent with the momentum budget (their interpretive framework). Much more work needs to be done regarding friction if it is to be used to interpret these simulations.

We will expand significatively the discussion of the potential role of friction modifying the ones initially proposed to be consistent with the momentum budget terms and including an additional estimate as explained in detail below after the specific comments of the reviewer on this point.

Without significant improvements in its main areas of new contributions, the work only marginally advances knowledge in the area of SAM persistence.

With this substantial revision and extended analyses, we consider that we can address the main criticisms of the reviewer and provide substantial advances on the role of high resolution (specifically of oceanic processes) on the simulated SAM persistence.

Finally, the work would be much stronger if it was better contextualized in the SAM feedback literature. It follows the SAM model bias literature reasonably well. Specifically, it should consider a few key areas which have important bearing on its results: 1) the evidence that the "feedback" which appears in austral summer (the focus of this work) is likely not due to eddy-mean flow interaction but nonstationary stratospheric variability (e.g., Byrne et al. 2016). 2) the understanding of SAM feedback mechanisms [barotropic (e.g. Lorenz and Hartmann 2001, Chen et al. 2008) vs baroclinic (e.g., Robinson 2000) vs diabatic (Xia and Chang 2014, Smith et al. 2024)] and how those feedback mechanisms might explain the role of surface friction and SST gradients. 3) The evidence for propagation of the SAM and for the stronger connection between SAM propagation biases and persistence biases than for eddy-feedback biases and persistence biases (e.g., Lubis and Hassanzadeh 2023). 4) the

importance of the SAM timescale for climate predictability (e.g., Simpson and Polvani 2016, Ma et al. 2017, Hassanzadeh and Kuang 2019).

 The main topic of our study is the ability of models to reproduce the observed SAM persistence and to determine if a better representation of meso-scale oceanic processes reduce the biases seen in lower resolution models. The introduction of our submitted manuscript was thus mainly devoted to those 'model-related' elements. However, we agree with the Reviewer that a longer discussion of the literature devoted to SAM persistence in general and specifically of the processes at the origin of this persistence would put our results in a broader context and justify more explicitly the choice of some of our analyses.

Consequently, we will add in the revised version a paragraph summarizing the main SAM feedback mechanisms. We will also come back to those mechanisms to discuss how mesoscale oceanic features could influence the SAM persistence (point first raised in another comment above). We will also add a sentence justifying the importance of the SAM persistence for predictability. This proposed paragraph is still relatively short considering the extensive literature on the subject.  Describing all the mechanisms at play and discussing the uncertainties of those mechanisms would require a lengthy introduction considering the goal of the paper, but we will come back to the relevant points in other sections of the manuscript, in particular in the conclusion when discussing some of the limitations of our study.

Proposed paragraph to replace the lines 55-58 of the submitted version where we were defining SAM persistence:

(*) 'A key characteristic of SAM is its temporal persistence, referring to how long a given phase of the SAM (positive, negative or neutral) tends to last before transitioning. This long persistence is important as it provides a source of predictability at a timescale longer than the one associated with synoptic variability (e.g., Robinson 2000; Lorenz and Hartmann 2001, Simpson and Polvani 2016). The persistence is often measured as the decorrelation timescale (e-folding timescale) which indicates the average duration over which the SAM index remains strongly correlated with its past values. A standard explanation to SAM persistence is the reinforcement of the westerly flow anomalies by the atmospheric eddy momentum flux generated by those changes in the mean flow. Several mechanisms can be at the origin of this eddy-mean flow feedback, including barotropic processes related to anomalous wave propagation and breaking  and baroclinic processes associated with eddy generation and enhanced baroclinicity in the lower troposphere in response to shift in the westerly flow (e.g., Robinson 2000, Lorenz and Hartmann 2001, Zurita-Gotor et al. 2014, Hassanzadeh and Kuang, 2019). The westerly flow anomalies also induce changes in the

diabatic heating and cooling due to latent heat release and cloud radiative effect that modify the temperature gradients, and potentially affecting SAM persistence (Xia and Chang 2014, Smith et al.2024, Vishny et al. 2024). In addition to this eddy-mean flow feedback, SAM persistence can have an origin from the stratosphere, which introduce some non-stationary forcing to SAM. The main influence is likely in late spring and summer at the time of the seasonal breakdown of the stratospheric vortex (Simpson et al. 2011, Byrne et al. 2016, Byrne et al. 2017, Saggioro and Shepherd 2019). Furthermore, interactions between a stationary mode and a propagating mode of the zonal variability could also affect SAM persistence (Lubis and Hassanzadeh 2021, Sheshadri and Plumb 2017, Smith et al. 2024).'

'Among all the processes that influence SAM persistence, one of the focuses here is the role of atmosphere-ocean exchanges. Surface heat fluxes modify the atmospheric temperature gradients, the boundary layer structure and thus diabatic heating of the atmospheric column as well as the low-level baroclinicity, which have both a demonstrated impact on SAM persistence (Xia and Chang 2014, Smith et al. 2024 Robinson 2000; Zurita-Gotor et al. 2014 ). Surface stress also plays a role as it tends to damp the westerly winds but also to enhance baroclinity and the baroclinic feedback (Robinson 2000, Zurita-Gotor 2014, Vishny et al. 2024). We will more specifically analyze the role of mesoscale oceanic features, which strongly impact the surface heat fluxes and surface stress in the Southern Ocean -a hotspot of mesoscale activity - (Frenger et al., 2013; Bishop et al. 2017) but whose potential influence on SAM persistence have been studied to a minimum to date.'

Other adjustments will be made to the introduction to avoid repetitions while including the new paragraphs.

**Specific Comments**

Line 29: Assertion "eddy feedback is a better indicator" needs more justification and/or more clarity (better in what way? In what circumstances?)

We propose to replace better indicator by 'useful indicator'.

Line 30-31: "These findings...offer insights": More specific language would be stronger (what insights?)

This sentence will be modified to be more specific.

Introduction: I think this discussion would be stronger if it included the significance of the persistence. As it stands, the section reviews the SAM and its significance for SH climate, what persistence is, some of its potential causes (non-stationarity is not discussed, see

following comment), its biases in GCMs, and some potential solutions for these biases. The problem is identified, but there exists a kind of motivational gap. The papers conclusions would be strengthened for unfamiliar readers if the significance of persistence was explicitly discussed.

As discussed above in response to the general comments, we will add a sentence in the introduction to mention the interest of persistence to predictability of Southern Hemisphere climate.

Lines 67-70, 79-95: An eddy-jet feedback is not the only possible source of persistence for the SAM. There is substantial literature published after the papers reviewed here which highlights the possibility for a "feedback" caused by non-stationarity induced by stratospheric forcing (Byrne et al. 2016, 2017, Saggioro and Shepherd 2019, etc.). This kind of forcing is especially important during early summer, the focus of this paper (see Byrne et al. 2016), when the stratospheric polar vortex breaks down. Thus, the feedback parameters computed here may not be responding to any internal tropospheric dynamics but the coupled troposphere-stratosphere system. The bias correction of Simpson et al. (2013a) suggests that this non-stationarity may not influence model biases, but it does not eliminate its possibility as one source of the feedback (Simpson et al. 2011 probably even supports this). Ma et al. 2017 further supports the notion of an eddy-feedback independent of non-stationarity, but a comprehensive discussion of this would improve the interpretation of the results.

As discussed above in response to the general comments, we will add a paragraph in the revised version explaining that eddy-jet feedback is not the only possible source of persistence. The potential impact for our evaluation of the eddy-jet feedback will be discussed in more detail in the methodology section (see below). A sentence will also be added in this paragraph on the specific impact of those processes on the biases on model persistence:

'However, a wrong magnitude of the eddy feedback is not the only possible source for the too long SAM persistence in models. In particular, biased troposphere-stratosphere interactions or a too strong interaction between the stationary and propagating mode of the zonal wind could also induce a too long persistence (Lubis and Hassanzadeh 2021, 2023).'

Line 143: Which segments of spin-up runs are retained? How much time is allowed for equilibration before using it for analysis? Given the importance of the stratosphere for the SAM and the time for its equilibration (~ 1 year), I would hope at least the first year is excluded from the analysis

For the revision, we have discarded all the spin-up simulations from our analysis to alleviate the potential impact from the non-stationarity of the simulation on the τ estimation. For the remaining EERIE control and historical simulations analyzed in this study, there is a 50-year spin-up time with the IFS-FESOM2 and ICON coupled models (following the design of HighResMIP) and a 200-year spin-up with the HadGEM3-GC5-EERIE model (following the design of CMIP6 DECK).

Line 148: I'm not fully convinced by this reasoning. In part, Byrne et al. (2016) show that non-stationarity does influence the calculation of eddy feedbacks, even given linear detrending. The spin-up simulations are certainly non-stationary, although that somewhat depends on whether some/how many of the initial years are omitted. The control simulations are likely not subject to this, but the historical simulations may be as well. Presumably they are more like reanalysis, but the point is that the differing periods may in fact influence the results beyond removing their climatological means (stationary or not). This non-stationary influence is notable in ERA5, where the bootstrapping Figure 2 shows substantially different decay timescales. In the case of NDJ, more likely influenced by non-stationarity, the two estimates are nearly non-overlapping. This is another argument that the analysis time period is not a trivial consideration. One way to partially address this concern is to bootstrap estimates for EERIE simulations, as done with ERA, particularly for simulations with few ensemble members (9km-AMIP in particular). Another concern is that the 10 years available for most simulations is not long enough to see strong convergence of the timescale, especially in coupled models (Gerber et al. 2008).

We mentioned in the submitted version that the proposed detrending and removal of the seasonal cycle did not remove all the effects of non-stationarity. This was likely too short. We will modify this paragraph explaining that the procedure we applied is standard, but we will insist more that non-stationarity implies serious limitations in our approach and how bootstrapping is a way to partially address this concern. As discussed above, we will also analyze longer simulations, discard the spin ups and apply the bootstrapping to most experiments (see for instance Fig. R2.1 above), reducing some of the limitations compared to the submitted version.

Lines 172-174: I would like more clarification about the choice to test the sensitivity of SAM persistence to different ocean mesoscale features. I do not understand the motivation very clearly. The zonal-mean, vertically-averaged zonal wind is a planetary-scale phenomenon, and while it is sensitive to ocean meso-scale features, I do not understand why it might be sensitive to one type over the other. The atmospheric eddies which power SAM and (potentially) its persistence are of a scale of 1000km, 10-100 times the scale of these features. While such temperature gradients can be important for lower-level baroclinicity

and the organization of convection, the large-scale drivers of SAM represent a further aggregation of these smaller scale dynamics. Indeed, there is currently no proposed mechanism (so far as I am aware) which argues that SAM should respond differently to these features. The idea that high-frequency SST gradients might strengthen the boundary layer heat flux, potentially enhancing boundary layer drag and strengthening the baroclinic feedback could be one argument, but it does not differentiate between eddies and fronts. In general, these results should be discussed in light of theories for baroclinic feedbacks on SAM persistence (Robinson 2000, Zurita-Gotor 2014, Zurita-Gotor et al. 2014). Diabatic feedbacks may also play a role here (Xia and Chang 2014, Smith et al. 2024).

We agree with the reviewer that there is no strong theoretical justification for separating the analysis by different ocean mesoscale features. The sensitivity experiments were originally designed to explore broader and under-studied questions under the EERIE project — specifically, to investigate "the relative importance of sharp SST gradients associated with ocean fronts and transient ocean eddies on the large-scale extratropical atmospheric circulation" (C. Roberts et al., 2024b). For simplicity, and to assess whether any major differences in the context of SAM persistence might warrant further investigation, we initially kept the features separated in the submitted version. However, we acknowledge that this separation may cause confusion, in the revised version we will present them as a single combined group. We will also develop in more details the physical processes that could explain a potential role of high-frequency SST gradients on SAM persistence in the introduction (see above)

Line 230: "for the same date in a calendar year". I think I know what this means, but more clarity would be better

Equation (1): y seems to be year, but it is not explicitly defined. The separation of t into d, y could be more clearly explained (see previous comment)

Those two points will be clarified in the revised version.

Line 249: As mentioned previously, this should be repeated for simulations with few ensemble members (5 or less).

We have performed 1,000 bootstrap resampling iterations to estimate the sampling uncertainty for all EERIE simulations presented, and the results—where applicable—will be included in the revised manuscript (Fig. 2).

Figure 1: Other reviewers and readers may disagree with me, but I think Figure 1 belongs as supplemental materials. The freed up PU (publication unit) could be used much more

effectively for other topics, some of which already mentioned, some to be mentioned. A very large majority of WCD readers interested in SAM and SAM timescale know what the pattern looks like, and if not, it is easily found. A more useful figure might be comparing the pattern across models. A similar argument is true for the timeseries. The raw timeseries is not relevant to the analysis being performed. Both are referenced once, only in passing. Panel c is more useful, but it is a visual explanation of e-folding time, which will be familiar to many readers, climate-oriented and not. Figure 1 could be more useful if it also depicted how the eddy feedback parameter (b) is calculated, as this is a more complex and less familiar calculation. Even with such an inclusion, I have a hard time justifying including Figure 1 in the main body of the text.

We agree that Figure 1 is relatively simple and does not present new findings beyond what is already available in the literature. However, based on previous interactions with researchers less familiar with SAM persistence, we have found that the concept of e-folding time is not always intuitive. Given that our manuscript contains a limited number of figures, we believe that including this introductory illustration may aid some readers without distracting more specialized audiences. To further justify its inclusion, we have added subplots to Figure 1 that illustrate how the eddy feedback strength is calculated (R2-2).

[Figure]

Figure R2-2. Example of the SAM decorrelation timescale and eddy feedback strength calculation based on

ERA5: (a) The first EOF pattern based on geopotential; (b) Associated first PC1 time series (only a partial segment is shown here); (c) Autocorrelation function (ACF) of the SAM index (smoothed with a Gaussian filter) shown for a given day of the year (black dashed), and an exponential fit (yellow). The e-folding timescale is denoted as τ. The ACF is repeated 1,000 times (gray) with the bootstrap sampling with replacement. (d) Same as (a) but based on vertically averaged zonal wind. (e) Lagged regression of [u]s and [m]s onto the SAM index. (f) Eddy feedback strength b for lags 7–14 days. Green for NDJ and black for annual-averaged results. Line 265: Because many of your models have different resolutions (particularly CMIP5 vs EERIE, you mention regridding CMIP5 to the same grid, but not EERIE), I would highly suggest following Menzel et al. (2019) or Barnes and Polvani (2015) and doing quadratic interpolation around the jet maximum to define the jet latitude. This will alleviate some of the degeneracy (models with identical jet latitudes) in Figure 3 and is consistent with the literature.

As suggested by the reviewer, we have performed a quadratic interpolation around the jet latitude maximum. We did not observe any major difference in the CMIP6 simulations. We will update the figures in the revised version, and add some clarifications in sec. 3.2:

L266: *The westerly jet is diagnosed following Menzel et al. (2019) and Barnes and Polvani (2015). We apply a quadratic fit method on the monthly mean zonally averaged 850-hPa zonal wind at the latitude where the maximum value is found between 75S and 10S and the four adjacent latitudes of the model. The latitude where the maximum value of the quadratic fit is found defines the position of the tropospheric westerly jet.*

Line 273: The switch from geopotential height for defining the timescale to zonal wind for defining the feedback is not without caveats. The assumption here is that the wind relevant for the SAM (and its feedbacks) is the geostrophic wind. Recently, however, Smith et al. (2024) demonstrate that SAM has significant eddy-feedbacks from the ageostrophic momentum fluxes which are leading-order in DJF in MERRA2. Vishny et al. (2024) also find important contributions to persistence from the ageostrophically-driven mean meridional circulation in idealized simulations. Thus, the imputation that models whose decay timescale is based on geopotential height will be consistent with the feedback from full (geostrophic+ageostrophic) zonal wind is probable, but not guaranteed. I think there is enough literature supporting the use of both (geopotential height and zonal wind) methods that it is not reasonable to redo the ACF calculations using zonal wind, but I do think it is worth acknowledging the geostrophic assumption and its limitations.

As suggested, we will add in the revised version a discussion of the limitations of the geostrophic assumption.

Line 278: I think the choice of three levels by Simpson et al. (2013b) was not intended to be the ideal, rather it was the best available information at the time (CMIP3). The vertical

structure of SAM can be quite nuanced despite its barotropic nature (Wall et al. 2022, Sheshadri et al. 2018), and a significant fraction of the eddy momentum flux necessary for the feedback exists above 250 hPa (Nie et al. 2014, Sheshadri et al. 2018). I suspect much more than those three levels are available for CMIP6, and there inclusion would strengthen this analysis.

We agree that including additional vertical levels would enhance the analysis. However, this choice was limited by computational constraints associated with the high-resolution EERIE simulations. Although this diagnostic focuses solely on the AMIP simulations, there are still a total of 18 simulations, encompassing three sensitivity experiments conducted at resolutions of 9 km and 28 km. The 28 km experiments, in particular, include five ensemble members each. As this is an international project that must balance the requirements of multiple teams within limited storage capacity, the 6-hourly data availability is currently restricted to only three vertical levels. To maintain consistency and enable comparison, we intentionally requested the same three levels used by Simpson et al. (2013b), allowing for a direct check against their results.

Line 300: The assumption of the Simpson framework is that the PCs are uncorrelated. Sheshadri and Plumb (2017), Lubis and Hassanzadeh (2020), Lubis and Hassanzadeh (2023), and Smith et al. (2024) have all shown this is not the case. Specifically, Sheshadri and Plumb (2017), Lubis and Hassanzadeh (2020), and Lubis and Hassanzadeh (2023) have shown that the coupling between EOF1 and EOF2 influences the SAM persistence timescale and the estimation of the eddy feedback parameter, and that the SAM timescale in CMIP6 models shows a strong dependence on the strength of the coupling between EOF1 and EOF2 (as measured by SAM's propagation period, see Lubis and Hassanzadeh 2023, Figure 7). Without examination of the coupling between modes across models, the spread in eddy-feedback parameters is difficult to interpret.

To address this comment, we will include the following paragraph in the revised version.

The approach followed here assumes that analyzing only the first PC is a good approximation to study SAM persistence. However, although the PCs are uncorrelated by construction on short timescale, this is not the case at longer lags and the coupling between the first two components influences SAM persistence (Sheshadri and Plumb 2017, Lubis and Hassanzadeh 2021, and Lubis and Hassanzadeh 2023). Analyzing only the first PC brings thus clear limitations in our analysis of the model spread in simulated SAM persistence. Furthermore, positive regression coefficients could be caused by non-stationarity of the series and in particular by interaction with the stratosphere and not just by eddy mean flow interactions. This introduces biases in the estimate of eddy feedback, particularly in late spring and summer (Byrne et al. 2016, Byrne et al. 2017), although this

does not necessarily prevent using the regression method (Ma et al. 2017). The methodology is thus imperfect, but it provides an interpretative framework for the difference between the simulations and allows a comparison with earlier studies. We will test here how useful it could be to analyze the behavior of the different modes, keeping in mind the possible caveats.

Line 310-325: I have two concerns involving the friction term. First, more could be done to properly estimate it and, second, utilize it in the interpretation of the results. I will begin with its estimation. Given $\tau$ as the surface stress, one can estimate the resulting torque as $d(\rho^{-1}\tau)/dz$ (see Vallis 2006, eq. 2.270), $\rho$ being density. If you only have $\tau$ at the surface, because you are vertically-integrating it, you can simply use the surface value and divide by the depth (in meters) of the atmospheric column, as the turbulent stress is likely zero at the model top. This faux-integration also yields a net negative sign (since the stress decreases with height), and should be of the right units ($N/m^2/kg*m^3/m=m/s^2$) and the correct sign. This approach should still be approximately valid in the case that the "surface" stress is actually the output turbulent stress from the boundary layer scheme for the full boundary layer.

The only information available about friction in the simulations is the surface stress. This is the reason why we use this quantity in our correlation analyses in the submitted version (Figure 4d-h). Nevertheless, we agree with the Reviewer that this quantity does not have the same unity as the terms in the momentum budget (equation 3) of the submitted version. This was mentioned as a note in line 293 of the submitted manuscript but performing the integration as suggested by the Reviewer is more elegant. We will thus follow the suggestion in the revised version. This will not change our correlations but will provide a term whose magnitude can be interpreted more easily. It is also more convenient for the comparison with other estimates of the friction term as explained below.

However, the friction in Lorenz and Hartmann (2001; and in other studies building on this framework) is generally parameterized as Rayleigh drag with a constant damping timescale. LH2001 explain in their Appendix A how to estimate it from timeseries of m and z. Since both of these fields are used in this analysis, it should be possible to estimate a friction via Rayleigh drag. This has two key benefits: 1) it can be used to validate the friction estimated from the stress, and triple checked against the residual of the momentum budget, evaluated from your equation (3), which should also be possible. In my experience, the residual usually matches the Rayleigh drag quite well. The second benefit is that it is useful for the interpretation of the feedback parameter, which I discuss more later.

Lorenz and Hartmann (2001) quantified the eddy feedback and the frictional term using spectral analysis and cross covariance. Here, we follow a different approach to estimate the feedback parameter, following Simpson et al. (2013) (see for instance the appendix of

Simpson et al. 2013b for a justification). Computing the frictional term following Lorenz and Hartmann (2001) may thus be confusing and would require additional technical justification in the manuscript. This would also introduce more complexity in our discussion and so we suggest not applying this methodology. By contrast, we will compute the friction term as a residual of the momentum budget, as suggested. This will allow us to double-check our results (using 2 of the 3 proposed methods). We can then obtain a damping timescale, equivalent to the one corresponding to a Raileigh drag, by performing the ratio between the zonal index (in m/s) and the estimated friction term (in m/s2).

A final issue with the estimation of the friction (no matter which method, preferably at least 2 of the 3) is that its projection value is proportional to the square root of the number of latitudes, and thus its magnitude should not be compared directly with simulations with different horizontal resolution. This is true for all the budget terms, but the feedback parameter is resolution-independent because it involves the ratio of two budget terms. To understand why, consider a simplified version of your equation (2) where **W** = **I** (the identity). If **e** is a (square-) integrable function f($\lambda$), sampled on an equally spaced grid (reasonable for GCM output), its Euclidean norm will be proportional to the square-root of the integral of [f($\lambda$)]$^2$ over latitude ($\lambda$), divided by the grid spacing (since we multiplied by the grid spacing to convert the sum into an integral). The integral should converge to the same value regardless of the resolution for most smoothly-varying, well-resolved f (again reasonable at even coarse GCM resolutions). However, the inverse of the grid spacing is proportional to the number of latitudes N (if the grid is evenly spaced). Thus the norm of **e** is proportional to $\sqrt{N}$. The multiplication of **Xe** is proportional to N (not the square root), by the same logic (because **e** is an orthogonal basis, the only component of **X** that survives the integration is proportional to **e**, and the product is proportional to **ee**). However, **Xe** has no square root, and thus **Xe**/$\sqrt{}$**ee** is proportional to $\sqrt{N}$. See a small example which should generalize well as the attached image. Note that including a non-identity weighting matrix **W≠I** does not change this, it simply adds another term into the integration. One could divide by $\sqrt{N}$ to alleviate this, or use integrals in the top and bottom instead. Or, one could divide the friction by the zonal wind projection as done for the feedback parameter. At that point, you may as well compute the damping timescale following the literature (LH2001, Appendix A).

We would like to thank the Reviewer for spotting this point. In the submitted version, all the results are interpolated on a ¼° grid before estimating the friction term (this will be stated explicitly in the revised version) so the different experiments can be compared, and the original grid size does not introduce a relative bias in this comparison. Nevertheless, as the issue remains for the absolute numbers, we will follow the suggestion to show the damping

timescale, which does not have this problem when estimated from the ratio of the zonal wind projection and the friction term projection.

The second friction-related issue is with the interpretation of the feedback. Following LH2001, the eddy feedback parameter (b) lengthens the effective timescale for the SAM by $t_f/(1-b*t_f)$, where $t_f$ is the frictional timescale. Thus, both the eddy feedback and the frictional timescale can effect SAM's persistence, and if models have differing frictional timescales, it could also explain differences in their persistence. In theory, one could see if this effective timescale $t_f/(1-b*t_f)$ followed the autocorrelation timescale more closely (I suspect it would), but the model bias literature (Gerber et al. 2008, Kidston and Gerber 2010) generally does not follow this convention, so I don't think this is strictly necessary. However, it may give a better interpretive framework for the friction to plot the frictional timescale (rather than the projection) and use this LH2001 relation to explain how the frictional timescale interacts with the eddy-feedback parameter to determine the total timescale.

As we will discuss the friction timescale in the revised version (see above), we can then test if combining estimates of friction timescale and eddy feedback as suggested provides a better interpretation of the difference in SAM persistence between the simulations.

Figure 2 (caption): Please describe the violin plot in more detail; I don't believe they are common enough to assume they can be interpreted properly without explanation.

We will modify the Figure caption to

"Figure R2-1. Distribution of τ (days) in CMIP6, EERIE coupled, and EERIE atmosphere-only (AMIP) simulations. CMIP6 and EERIE AMIP are both historical simulations, with a fixed period indicated in the x-axis labels, and the EERIE coupled simulations cover varied periods as indicated in Table 1. ERA5 is analyzed for two time periods for references. **CMIP6 results from 31 experiments are presented in violin plot, in which the width indicates the density of the data points, the thin gray vertical box in the middle shows the 25th –75th quantiles, and the white dot presents the median.** For the rest, error bars are added wherever applicable to show the ±1 standard deviation of τ from the 1,000 bootstrap resamples."

Lines 396-398: I think this point on the interpretation of the IFS-AMIP experiments requires more discussion and computation. These are an IC ensemble from the same GCM with the same boundary conditions, and thus represent internal variability of the same mean climate in a way that isn't the case for comparisons across the CMIP models. For example, you could likely run 5 more IC ensembles, and you might get a completely different pattern between

jet latitude and e-folding timescale. But I don't think that somehow contradicts the expectation that the two should be positively correlated due to the stronger wave reflection (and weaker feedback) of more poleward jets (Barnes and Hartmann 2010, Lorenz 2023). Despite this, according to the convergence estimates of Gerber et al. (2008), the 40 years of AMIP simulations should be enough to constrain the decorrelation timescale within a day, and 4 of the ensemble members are within one day of their mean. This is where I think bootstrapped estimates of the sampling uncertainty could help resolve this question of whether sampling uncertainty can explain the lack of relationship, or whether this is indeed a breakdown of the expected theory.

We agree with the Reviewer that the IC ensemble is different from CMIP models as the boundary conditions are the same for all the IC members while ensemble of coupled models can have very different SST patterns for instance. This will be made more explicit in the revised version by mentioning 'internal atmospheric variability' instead of just internal variability'. We also agree that there is no reason to consider that this difference contradicts the expectation that jet latitude and e-folding time should be positively correlated. One hypothesis we put forward is that, when the position of the jet is well captured (as in all IFS-AMIP experiments), the difference in jet position is too small between the different experiments (compared for instance with CMIP models) to explain the difference in decorrelation timescale and other factors dominates, but this is only a speculation at this stage. We will modify this paragraph in the revised version to make this clearer, we will also check how the bootstrapping influences the robustness of our conclusions and insist that the 'documented bias relationship between τ and λ0 in the literature' is based on strong physical arguments.

Figure 3: When uncertainty exists in both the dependent and independent variables of a regression, it may be more appropriate to use a different type of regression than least-squares, especially if the uncertainties are correlated (see Pendergrass and Kao 2022, and York 2004 for an alternative).

We agree that a more sophisticated regression method could be more precise, but we have chosen here the standard least-squares for simplicity as in some previous studies.

Line 415: Sample size of one, not enough evidence to support conclusion (bootstrapping would help)

We will add the standard deviation of τ for all AMIP simulations using bootstrap resampling. The decrease in τ with higher resolution is not particularly robust on the annual-mean scale, as the estimated τ in the 9-km simulation falls within (though toward the lower end of) the

range covered by the 28-km simulations (Fig. 2a). However, during the NDJ season, the reduction in τ for the 9-km simulation is more pronounced, with its estimated range extended outside those of the 28-km simulations (Fig. 2b).

Lines 425-430: Some connection to existing feedback mechanisms would be appropriate here

We will add some discussion on such connection in the revised manuscript.

Lines 441-449: See previous comments regarding friction

This section will be updated using the new results. In particular, we will add references to feedback mechanisms and more details regarding friction.

Line 462: Is the convection parameterization turned off at 9km? Stronger latent heating in the 9km run could create a stronger negative diabatic feedback (Xia and Chang 2014), decreasing the persistence

The convection parameterization is still active at 9 km, as at 27 km resolution, consistently with the convection parameterization settings applied in ECMWF operational forecasts.

Figure 4: Panels (f), (g) and (h) should be greatly simplified, maybe down to one panel (or even a table), showing the simulation on one axis and the value of the x axis on the other. The decay timescales are identical, and two points is not enough to infer any relationship, so the current scatter plots visually complicate the comparison between simulations. Readers will understand why they do not follow (b), (c), and (d), no need to artificially fit that pattern.

Following the reviewer's suggestions, we will remove the panels e-f from the submitted version and combine the results of the 9-km members to the 28-km members (originally shown on the panels b-d). An example figure is shown here as Fig. R2-3, but we will update it with the new results in the revised manuscript.

[Figure]

Figure R2-3. (a) SAM decorrelation timescale (τ) as a function of month for IFS-AMIP 28km simulations (dashed lines for ensemble members and solid lines for the ensemble means; experiments indicated in the legend) and ERA5 (yellow). (b) Same as (a) but for 9 km. (c) Scatter plot of τ (days) and westerly jet latitude (filled-color markers are for 28 km members, and the unfilled stars are for 9 km simulations). (d)–(e) Similar to (c) but for the eddy feedback strength and frictional effects associated with SAM, respectively. In (b)–(d), the gray dotted line represents the linear regression fit, and the correlation coefficient and p value are indicated in the top-right corner.

Line 492: For the 10 year, coupled EERIE simulations, I'm not convinced this is long enough to really reduce the sampling uncertainty, which converges very slowly (see Gerber et al. 2008). Bootstrapped measures would help alleviate this concern; without such attempts, it is hard to interpret the difference between the EERIE simulations and the longer CMIP6 simulations (and the longer IFS-AMIP simulations for that matter).

As our responses above, we will analyze extended simulations (65 years with IFS-FESOM2, 30 years with HadGEM3, and 22 years with ICON) and apply bootstrapping resampling to strengthen our analysis.

Line 540: please clarify: better indicator of what?

A better indicator of SAM persistence. This will be specified in the revised version (see the response to the next comment).

Line 541: "more statistically significantly" If I recall, while the p-value was small, the result was not significant. I think significance is too binary for this language. I would stick with language which discusses what a small p-value means (the relationship is unlikely to be due to random chance).

The lower the p-value is, the lower the probability of getting that result if the null hypothesis were true. The null hypothesis here is that there is no correlation between the two variables. Therefore, with a smaller p, we can reject the null hypothesis and conclude that there is a linear relationship between the two variables with a higher confidence. We will rephrase the lines from

"It is worth noting that the eddy feedback strength appears to be a better indicator than the mean-state jet latitude $\lambda_0$, linking positively and more statistically significantly to the simulated summertime $\tau$ among the IFS AMIP simulations."

to

"It is worth noting that, compared to the mean-state jet latitude ($\lambda_0$), the eddy feedback strength shows a stronger correlation with early-summer $\tau$, as indicated by a larger magnitude of the Pearson correlation coefficient and a lower p-value (i.e., higher statistical confidence) across the IFS AMIP simulations. "

Line 522: I'm not convinced the path forward is that promising from these results. A higher resolution atmosphere helps. That is good. But it does not seem to benefit from being coupled (bias improves in AMIP) and it does not seem to benefit from mesoscale ocean features (smoothed SST runs have lower bias). Improvements in jet latitude at these resolutions do not seem to help either. However, the climate community will want to run coupled models for the estimation of climate variability and sensitivity for the foreseeable future. If other models behave like IFS (a big assumption), it is likely models will be stuck with some irreducible bias in SAM timescale. Perhaps I am too pessimistic. If so, please help me understand what other path these results suggest.

Suggesting perspectives at the end of a study is always a difficult task and bears many uncertainties. Based on the Reviewer's comments and of the new analysis carried out, we will modify this paragraph to be more explicit about the main implications of our study. We agree that the AMIP simulations seem to indicate that two-way coupling between the atmosphere and the ocean or the mesoscale eddies are not essential to reproduce well SAM persistence. On the other hand, those experiences emphasized that a good representation of sea surface temperature is critical. Many oceanic studies have demonstrated the key role of mesoscale eddies in the Southern Ocean and that eddy-permitting models often have

larger SST biases than either models with parameterized eddy fluxes or eddy-rich models (e.g. Storkey et al. 2025). The impact of mesoscale eddies may be only indirect but still essential.

**Technical Corrections**

Lines 73, 90, 240: Simpson 2013 referenced without a/b

Lines 376-379: This sentence "However, it is also possible… more critical" could benefit from more clarity, including maybe breaking into smaller sentences.

Lines 511-514: Rephrasing (and separating into smaller sentences) would improve clarity here

The suggested technical corrections will be implemented in the revised version.

**References**

Barnes, E. A., & Hartmann, D. L. (2010). Testing a theory for the effect of latitude on the persistence of eddy-driven jets using CMIP3 simulations. Geophysical Research Letters, 37(15). https://doi.org/10.1029/2010GL044144

Barnes, E. A., & Polvani, L. M. (2015). CMIP5 Projections of Arctic Amplification, of the North American/North Atlantic Circulation, and of Their Relationship. Journal of Climate, 28(13), 5254–5271. https://doi.org/10.1175/JCLI-D-14-00589.1

Bishop, S. P., Small, R. J., Bryan, F. O. & Tomas, R. A. Scale dependence of midlatitude air–sea interaction. J. Clim. 30, 8207–8221. https:// doi. org/ 10. 1175/ JCLI-D- 17- 0159.1 (2017).

Byrne, N. J., Shepherd, T. G., Woollings, T., & Plumb, R. A. (2016). Annular modes and apparent eddy feedbacks in the Southern Hemisphere. Geophysical Research Letters, 43(8), 3897–3902. https://doi.org/10.1002/2016GL068851

Byrne, N. J., Shepherd, T. G., Woollings, T., & Plumb, R. A. (2017). Nonstationarity in Southern Hemisphere Climate Variability Associated with the Seasonal Breakdown of the Stratospheric Polar Vortex. Journal of Climate, 30(18), 7125–7139. https://doi.org/10.1175/JCLI-D-17-0097.1

Chen, G., Lu, J., & Frierson, D. M. W. (2008). Phase Speed Spectra and the Latitude of Surface Westerlies: Interannual Variability and Global Warming Trend. Journal of Climate, 21(22), 5942–5959. https://doi.org/10.1175/2008JCLI2306.1

Frenger, I. et al., 2013. Imprint of Southern Ocean eddies on winds, clouds and rainfall. Nature Geoscience 6, 608–612.

Gerber, E. P., Voronin, S., & Polvani, L. M. (2008). Testing the Annular Mode Autocorrelation Time Scale in Simple Atmospheric General Circulation Models. Monthly Weather Review, 136(4), 1523–1536. https://doi.org/10.1175/2007MWR2211.1

Hassanzadeh, P., & Kuang, Z. (2019). Quantifying the Annular Mode Dynamics in an Idealized Atmosphere. Journal of the Atmospheric Sciences, 76(4), 1107–1124. https://doi.org/10.1175/jas-d-18-0268.1

Kidston, J., & Gerber, E. P. (2010). Intermodel variability of the poleward shift of the austral jet stream in the CMIP3 integrations linked to biases in 20th century climatology. Geophysical Research Letters, 37(9). https://doi.org/10.1029/2010GL042873

Lorenz, D. J. (2023). A Simple Mechanistic Model of Wave–Mean Flow Feedbacks, Poleward Jet Shifts, and the Annular Mode. Journal of the Atmospheric Sciences, 80(2), 549–568. https://doi.org/10.1175/JAS-D-22-0056.1

Lorenz, D. J., & Hartmann, D. L. (2001). Eddy–Zonal Flow Feedback in the Southern Hemisphere. Journal of the Atmospheric Sciences, 58(21), 3312–3327. https://doi.org/10.1175/1520-0469(2001)058<3312:EZFFIT>2.0.CO;2

Lubis, S. W., & Hassanzadeh, P. (2020). An Eddy–Zonal Flow Feedback Model for Propagating Annular Modes. Journal of the Atmospheric Sciences, 78(1), 249–267. https://doi.org/10.1175/JAS-D-20-0214.1

Lubis, S. W., & Hassanzadeh, P. (2023). The Intrinsic 150-Day Periodicity of the Southern Hemisphere Extratropical Large-Scale Atmospheric Circulation. AGU Advances, 4(3), e2022AV000833. https://doi.org/10.1029/2022AV000833

Ma, D., Hassanzadeh, P., & Kuang, Z. (2017). Quantifying the Eddy–Jet Feedback Strength of the Annular Mode in an Idealized GCM and Reanalysis Data. Journal of the Atmospheric Sciences, 74(2), 393–407. https://doi.org/10.1175/JAS-D-16-0157.1

Menzel, M. E., Waugh, D., & Grise, K. (2019). Disconnect Between Hadley Cell and Subtropical Jet Variability and Response to Increased CO2. Geophysical Research Letters, 46(12), 7045–7053. https://doi.org/10.1029/2019GL083345

Nie, Y., Zhang, Y., Chen, G., Yang, X.-Q., & Burrows, D. A. (2014). Quantifying barotropic and baroclinic eddy feedbacks in the persistence of the Southern Annular Mode. Geophysical Research Letters, 41(23), 8636–8644. https://doi.org/10.1002/2014GL062210

Roberts, C., Aengenheyster, M., Roberts, M., 2024b. Description of EERIE idealised atmosphere-only simulations. https://doi.org/10.5281/ZENODO.14514510

Robinson, W. A. (2000). A Baroclinic Mechanism for the Eddy Feedback on the Zonal Index. Journal of the Atmospheric Sciences, 57(3), 415–422. https://doi.org/10.1175/1520-0469(2000)057<0415:ABMFTE>2.0.CO;2

Saggioro, E., & Shepherd, T. G. (2019). Quantifying the Timescale and Strength of Southern Hemisphere Intraseasonal Stratosphere-troposphere Coupling. Geophysical Research Letters, 46(22), 13479–13487. https://doi.org/10.1029/2019GL084763

Sheshadri, A., & Plumb, R. A. (2017). Propagating Annular Modes: Empirical Orthogonal Functions, Principal Oscillation Patterns, and Time Scales. Journal of the Atmospheric Sciences, 74(5), 1345–1361. https://doi.org/10.1175/JAS-D-16-0291.1

Sheshadri, A., Plumb, R. A., Lindgren, E. A., & Domeisen, D. I. V. (2018). The Vertical Structure of Annular Modes. Journal of the Atmospheric Sciences, 75(10), 3507–3519. https://doi.org/10.1175/JAS-D-17-0399.1

Simpson, I. R., Hitchcock, P., Shepherd, T. G., & Scinocca, J. F. (2011). Stratospheric variability and tropospheric annular-mode timescales. Geophysical Research Letters, 38(20). https://doi.org/10.1029/2011GL049304

Simpson, Isla R., & Polvani, L. M. (2016). Revisiting the relationship between jet position, forced response, and annular mode variability in the southern midlatitudes. Geophysical Research Letters, 43(6), 2896–2903. https://doi.org/10.1002/2016GL067989

Simpson, Isla R., Hitchcock, P., Shepherd, T. G., & Scinocca, J. F. (2013). Southern Annular Mode Dynamics in Observations and Models. Part I: The Influence of Climatological Zonal Wind Biases in a Comprehensive GCM. Journal of Climate, 26(11), 3953–3967. https://doi.org/10.1175/JCLI-D-12-00348.1

Simpson, Isla R., Shepherd, T. G., Hitchcock, P., & Scinocca, J. F. (2013). Southern Annular Mode Dynamics in Observations and Models. Part II: Eddy Feedbacks. Journal of Climate, 26(14), 5220–5241. https://doi.org/10.1175/JCLI-D-12-00495.1

Smith, S., Lu, J., & Staten, P. W. (2024). Diabatic Eddy Forcing Increases Persistence and Opposes Propagation of the Southern Annular Mode in MERRA-2. Journal of the Atmospheric Sciences, 81(4), 743–764. https://doi.org/10.1175/JAS-D-23-0019.1

Storkey D, P. Mathiot, M. J. Bell, D. Copsey, C. Guiavarc'h, H. T. Hewitt, J. Ridley, and Ma. J. Roberts (2025). Resolution dependence of interlinked Southern Ocean biases in global coupled HadGEM3 models.  Geosci. Model Dev., 18, 2725–2745, 2025. https://doi.org/10.5194/gmd-18-2725-2025

Vallis, G. K. (2006). Atmospheric and Oceanic Fluid Dynamics (1st ed.). Cambridge University Press.

Vishny, D. N., Wall, C. J., & Lutsko, N. J. (2024). Impact of Atmospheric Cloud Radiative Effects on Annular Mode Persistence in Idealized Simulations. Geophysical Research Letters, 51(15), e2024GL109420. https://doi.org/10.1029/2024GL109420

Wall, C. J., Lutsko, N. J., & Vishny, D. N. (2022). Revisiting Cloud Radiative Heating and the Southern Annular Mode. Geophysical Research Letters, 49(19), e2022GL100463. https://doi.org/10.1029/2022GL100463

Xia, X., & Chang, E. K. M. (2014). Diabatic Damping of Zonal Index Variations. Journal of the Atmospheric Sciences, 71(8), 3090–3105. https://doi.org/10.1175/JAS-D-13-0292.1

York, D., Evensen, N. M., Martínez, M. L., & De Basabe Delgado, J. (2004). Unified equations for the slope, intercept, and standard errors of the best straight line. American Journal of Physics, 72(3), 367–375. https://doi.org/10.1119/1.1632486

Zurita-Gotor, P. (2014). On the Sensitivity of Zonal-Index Persistence to Friction. Journal of the Atmospheric Sciences, 71(10), 3788–3800. https://doi.org/10.1175/JAS-D-14-0067.1

Zurita-Gotor, P., Blanco-Fuentes, J., & Gerber, E. P. (2014). The Impact of Baroclinic Eddy Feedback on the Persistence of Jet Variability in the Two-Layer Model. Journal of the Atmospheric Sciences, 71(1), 410–429. https://doi.org/10.1175/JAS-D-13-0102.1

---

## Author Response (AR1)

Authors' Response to referees' comments on "Southern Annular Mode Persistence and

Westerly Jet: A Reassessment Using High-Resolution Global Models" by Chen et al.

MS No.: egusphere-2025-666 MS type: Research article

Following the reviewers' advice, we have made substantial efforts to address their concerns—particularly those raised by Reviewer 2—and revised the manuscript accordingly. In summary, we have:

- Included a supplementary table presenting the results for each individual CMIP6 model.
- Replaced the EERIE spin-up runs and short simulations with longer (22–65 year) runs, including two complete historical simulations, and added bootstrapping.
- Expanded the literature review with a broader discussion of processes affecting SAM
  persistence and clarified the motivation for the AMIP (IFS) sensitivity experiments with
  varied SST boundary conditions.
- Implemented two of the reviewer 2's suggested methods for estimating frictional impacts and expanded the related discussion.
- Adopted a reviewer-recommended method for identifying the westerly jet location (though results remain similar).

Additional revisions have been made to the article to reduce repetition and incorporate new content. All figures have been improved compared to their previous versions. Detailed responses to each comment are provided below, with the reviewer's comments in black and our responses in green. The line numbers refer to the revised manuscript (without tracked changes); however, a tracked-changes version is also provided for reference.

**Referee 1 (RC1)**

**Overall Assessment**

This study explores the atmospheric and oceanic influences on modelled SAM persistence and its relationship with the latitude of the mid-latitude jet. The authors note the longstanding issue that CMIP models (including the latest suite: CMIP6) overestimate SAM persistence (quantified using the decorrelation timescale), particularly in early austral summer, which appears to be much improved when using high-resolution, eddy-resolving simulations from the EERIE project. This appears to be in part due to more realistic simulation of the jet position/distribution (CMIP models have tended to be too equatorward biased) but the importance of accurate SST representation is also clear. In fact, the authors show that AMIP model experiments of the EERIE simulations perform better than coupled experiments in terms of more realistically representing the jet position and SAM decorrelation timescale. Enhanced resolution of the EERIE simulations likely plays a role in the improvement relative to CMIP6 models but also improved model physics, particularly concerning ocean mesoscale eddies which appear to slightly enhance the SAM decorrelation timescale in early summer (at least for simulations run at 28 km resolution) according to sensitivity simulations performed. However, cancellation effects (e.g., atmospheric eddy feedback strength versus surface friction) make it difficult to ascertain which aspects help improve modelling of the SAM persistence. For instance, the decorrelation timescale is more realistic still in 9 versus 28 km AMIP simulations, yet the role of ocean mesoscale eddies in enhancing SAM persistence is not evident at this finer resolution. I found the study to be very well written, organised and logically structured. The Figures are clearly presented and straightforward to understand and the step-bystep computation of the different diagnostics examined will I think be much appreciated by many readers. The conclusions drawn are supported by the results shown, so I can recommend this be accepted for publication in Weather and Climate Dynamics. I include just a few comments for the authors to consider prior to acceptance.

**General Comments**

I only had one main consideration for the authors which I found lacking in the paper. That is information of which CMIP6 models were considered (if supplementary information were to be provided, a table for this would be warranted). That is not to say that knowledge of which models lie where in the distributions shown (Figures 2 and 3) are important in understanding this work. But for others reading, it might be useful for them to know and the best way I feel to include this would be to show similar Figures with individual CMIP6 models indicated in supplementary information. Nevertheless, knowledge of which CMIP6 models lie where could help others or even the authors to comment upon whether commonalities such as shared model components or known biases more widely within the climate system might influence the results. So, I would encourage the authors to think about providing this information.

We would like to thank the reviewer for their encouraging comments, and we have taken the suggestions into account for the revised version of the manuscript. A table has been added in the supplementary materials:

Table S1. Annual and early-summer (NDJ) SAM persistence (the decorrelation timescale of the SAM index;  $\tau$ ) and westerly jet position for the 31 studied CMIP6 models (historical simulations) and ERA5 for the same period of 1980-2014.

| Model           | τ Annual | τNDJ | Jet latitude Annual | Jet latitude NDJ |
|-----------------|----------|------|---------------------|------------------|
| TaiESM1         | 8.2      | 11.6 | -52.4               | -50.4            |
| AWI-ESM-1-1-LR  | 12.0     | 21.2 | -49.0               | -48.4            |
| AWI-ESM-1-REcoM | 14.1     | 25.1 | -49.7               | -50.4            |
| BCC-CSM2-MR     | 11.2     | 21.5 | -51.0               | -50.3            |
| BCC-ESM1        | 14.2     | 30.2 | -50.5               | -49.5            |
| FGOALS-f3-L     | 11.4     | 16.6 | -49.7               | -48.9            |
| FGOALS-g3       | 9.4      | 12.2 | -49.7               | -49.5            |
| CanESM5         | 10.2     | 14.5 | -49.6               | -49.8            |
| IITM-ESM        | 13.2     | 22.5 | -47.0               | -47.0            |
| CNRM-CM6-1      | 11.3     | 18.7 | -48.2               | -48.4            |
| CNRM-CM6-1-HR   | 9.8      | 12.6 | -45.8               | -46.2            |
| CNRM-ESM2-1     | 11.9     | 20.2 | -48.5               | -48.4            |
| ACCESS-CM2      | 13.9     | 27.0 | -50.1               | -49.6            |
| EC-Earth3       | 9.4      | 12.9 | -50.3               | -49.4            |
| MPI-ESM-1-2-HAM | 12.7     | 24.3 | -49.9               | -49.7            |
| INM-CM4-8       | 9.2      | 9.5  | -51.7               | -53.4            |
| INM-CM5-0       | 10.7     | 15.2 | -50.3               | -52.0            |
| IPSL-CM6A-LR    | 12.7     | 22.9 | -49.2               | -48.8            |
| MIROC6          | 11.7     | 20.3 | -48.7               | -49.6            |

| MPI-ESM1-2-HR            | 11.9        | 15.6         | -48.8          | -48.4          |
|--------------------------|-------------|--------------|----------------|----------------|
| MPI-ESM1-2-LR            | 10.0        | 14.3         | -47.9          | -48.4          |
| MRI-ESM2-0               | 16.0        | 32.1         | -48.0          | -47.0          |
| GISS-E2-1-G (1)          | 13.2        | 23.4         | -51.1          | -50.9          |
| GISS-E2-1-G (2)
CESM2 | 10.8
9.0 | 17.0
13.4 | -50.8
-51.9 | -50.5
-52.3 |
| CESM2-FV2                | 9.3         | 13.9         | -52.7          | -52.2          |
| CESM2-WACCM              | 11.6        | 16.3         | -52.0          | -51.9          |
| CESM2-WACCM-FV2          | 11.2        | 17.3         | -52.3          | -51.7          |
| NorESM2-LM               | 7.1         | 9.6          | -52.6          | -51.6          |
| NorESM2-MM               | 9.1         | 13.0         | -52.6          | -52.2          |
| GFDL-CM4                 | 9.2         | 12.4         | -49.0          | -48.7          |
| ERA5                     | 7.9         | 10.9         | -51.1          | -50.4          |

**Specific Comments**

L15: "a critical driver" → "a leading mode"? I wouldn't consider the SAM to a driver itself, but more of a reflection of driving influences. Expressing it as a leading mode would be more scientifically accurate and consistent with earlier literature (e.g., Marshall, 2003; Marshall et al., 2022).

We thank the Reviewer and agree with this comment. We have changed the sentence as suggested. (L15 in the revised manuscript).

L340-342: How reliable is the SAM derived from ERA5 pre-satellite era when comparing with the EERIE coupled simulations? Presumably the SAM is more reliably reconstructed after 1979 from ERA5 but maybe difficult to quantify how much of an improvement there would be. It may be worthy of further comment or caveating, however?

We agree that ERA5 is less reliable in the pre-satellite era (added to the Results section):

L366: "Note, however, that there is relatively less confidence in the accuracy of the value of the SAM in ERA5 prior to the satellite era."

But we have also added that ERA5 is still a relatively better option among the available datasets (in the Data section):

L216: "Among the reanalysis products that extend backwards in time beyond 1979 (ERA5, 20CRv3, JRA-55), ERA5 is found to agree best with station observations and produces good representation of SAM, both before and after the advent of satellite sounder data (Marshall et al., 2022)."

L343-344: Suggesting that a relative minority of models are considerably worse in representing SAM decorrelation timescale than the rest of the pack. Did the authors investigate why this might be or were they at least able to note some commonalities in the most unrealistic models that might point to the cause(s)? For instance, could there be an association between too equatorward jet position and shared model components? Or factors that may give plausibly give rise to the issue of realistic eddy feedback strength? It may be beyond the scope of the paper to delve into this, but others reading might be encouraged to look into this.

It is indeed something interesting to study. However, we did not identify obvious commonalities shared by the worse performing models (e.g. spatial resolution or warm bias over Antarctica or the Southern Ocean or shared components).

Previous studies using CMIP models have discussed the association between too equatorward jet position and models' issues to represent the SAM persistence (e.g., Bracegirdle 2020, Zhang 2021) and the related role of the eddy feedback strength. However, we have the feeling that a comprehensive study of the shared components of these models falls beyond the scope of the study, which focuses on the added value of the high-resolution EERIE models.

L477: "...a key driver..." → Again I think "...a leading mode..." would be more technically correct.

We have modified the sentence in the revised version accordingly:

L494-496: This study assesses the performance of new high-resolution global model simulations developed under the EERIE project in capturing the persistence of the Southern Annular Mode (SAM), a leading mode of climate variability in the Southern Hemisphere.

**Technical Corrections**

L52-53: "spring (MAM) and summer (JJA)". → "autumn (MAM) and winter (JJA)".

Corrected (L52-53). Thank you.

L153: "observation" → "observations" Table 1: Some font size inconsistencies noted.

This sentence has been removed from the revised manuscript. Font size of Table 1 has been adjusted.

L245: Tabulation before "Finally..."

Corrected (L259). Thanks.

**References**

Bracegirdle, T. J., Holmes, C. R., Hosking, J. S., Marshall, G. J., Osman, M., Patterson, M., & Rackow, T. (2020). Improvements in circumpolar Southern Hemisphere extratropical atmospheric circulation in CMIP6 compared to CMIP5. *Earth and Space Science*, 7(6), e2019EA001065.

Marshall, G. J. (2003). Trends in the Southern Annular Mode from observations and reanalyses. Journal of climate, 16(24), 4134-4143, https://doi.org/10.1175/15200442(2003)016<4134:TITSAM>2.0.CO;2.

Marshall, G. J., Fogt, R. L., Turner, J., & Clem, K. R. (2022). Can current reanalyses accurately portray changes in Southern Annular Mode structure prior to 1979?. *Climate Dynamics*, 59(11), 3717-3740, https://doi.org/10.1007/s00382-022-06292-3.

Zhang, X., He, B., Liu, Y., Bao, Q., Zheng, F., Li, J., ... & Wu, G. (2022). Evaluation of the seasonality and spatial aspects of the Southern Annular Mode in CMIP6 models. *International Journal of Climatology*, *42*(7), 3820-3837.

**Referee 2 (RC2)**

"Southern Annular Mode Persistence and the Westerly Jet: A Reassessment Using High-Resolution Models" examines the relationship between SAM persistence, the climatological jet latitude, and the classic eddy-feedback parameter in both CMIP6 models and a new suite of high-ocean-resolution models (EERIE), with some added AMIP-style simulations with one EERIE model to help interpret the effects of increased resolution. The work finds that EERIE simulations have much lower bias in the SAM timescale than CMIP6, particularly in summer (traditionally the worst season). The bias is even lower in the AMIP simulations forced by observational SSTs, which suggests that ocean-atmospheric coupling may contribute to the bias. While these EERIE simulations have a lower bias and lower resolution than most CMIP6 models, the CMIP models do not show much dependence on horizontal resolution. Instead, previously established relationships relating the SAM timescale to the jet latitude seem to hold for the CMIP models. For EERIE models, this relationship breaks down, and the eddy-feedback parameter has better correlations with the annular mode timescale. When SST gradients are reduced in the AMIP style simulations, the persistence is reduced, although the cause is unclear.

I cannot recommend the paper to be published in its current form. With substantial revision and extended analysis, it could eventually be published, but the current state of the paper presents only a very marginal advancement in knowledge in the area of SAM timescales, and the results are challenging to interpret without more context in the literature and clearer interpretive frameworks.

Despite these criticisms, the paper does a few things well. First, I think the question is well-defined: what are the impacts of high-resolution atmosphere and ocean models, and their coupling, on SAM persistence? I also think they have the data available to address this question, but it needs to be much better utilized. They outline their methodology in a very reproducible way, and generally they follow the previous literature (to a point). The writing is of good quality and reasonably easy to follow.

My major concerns are summarized below; a detailed discussion follows. The novel contributions of this work are the analysis of high-resolution simulations, the SST sensitivity experiments, and the consideration of friction to explain intermodel differences. All of these contributions require serious improvement.

We would like to thank the reviewer for the detailed and constructive evaluation of our manuscript, and we have made substantial suggested revisions to the manuscript. The additional material broadens the scope of our study, which was initially focused on a mainly descriptive, standard evaluation of the SAM in a new set of high-resolution simulations. The reviewer underlines three novel contributions in our work and that all three require serious improvement. The way we have introduced those improvements is explained in detail following each specific comment. The main changes are briefly summed up here:

- We have discarded spin-up EERIE simulations and replaced the shorter simulations with longer runs (ranging from 22-year to 65-year long, including 2 historical simulations) in our new analysis; and the sampling uncertainty in estimating their SAM e-folding timescale have been supplemented with the bootstrapping method.
- We have expanded the literature review and strengthened its connection to our results. This
  includes a broader discussion of the processes contributing to SAM persistence and an
  extended examination of the potential role of ocean mesoscales, providing a clearer
  justification for our AMIP sensitivity experiments using varied SST boundary conditions.
- We have followed two of the reviewer's suggestions to modify our approaches to estimate frictional impact on SAM persistence and expanded the relevant discussion.
- We have adopted the suggested papers' approach in identifying the jet location.

We also acknowledge that we cannot answer all the questions in a single study, but when it is not possible to have robust conclusions, we have highlighted it as well as the limitations of our study.

1. Regarding the analysis of high-resolution simulations: the simulations are all short (10 years) and frequently non-stationary (spin-up) or non-overlapping with the observational record. Given the long timescales required for SAM timescale convergence, the significant impacts of non-stationarity on the estimation of the timescale, and the potential for decadal and supra-decadal variability in the feedback itself (following the jet latitude), interpreting the difference between the EERIE simulations, ERA5, and CMIP6 is very challenging. Clearly the bias is reduced, but it is not clear at present whether this is due to artefacts, random chance, or physically meaningful reductions. This problem could be partially alleviated by carrying out the bootstrapping techniques used for the reanalysis for the EERIE simulations. Longer runs/overlapping time periods would be preferable, but given the computational expense involved the current simulations might be acceptable given appropriate explanation of the caveats involved.

Since the submission of our manuscript, new simulations have been available. This allows us to analyze longer runs and to discard all the spin-up simulations that may have strong issues with stationarity. The changes of the simulations are highlighted in the table below (see Table 1 in the revised manuscript):

| Institution | Alfred Wegener
Institute (AWI)           | Max Planck Institute
(MPI-M) | Met Office
(MO) | ECMWF             |
|-------------|---------------------------------------------|---------------------------------|--------------------|-------------------|
|             | Coupled atmosphere-ocean models (eddy-rich) |                                 |                    | Atmospheric model |

| System name                                     | IFS-FESOM2                                                            | ICON                                           | HadGEM3-
GC5-EERIE | IFS                       |
|-------------------------------------------------|-----------------------------------------------------------------------|------------------------------------------------|-----------------------|---------------------------|
| Simulations analyzed in initial submission      | 1950spinup (31 yrs)
1950control (20 yrs)
Historical (1950–1969) | 1950spinup
Cycle 2 (11 yrs)                 | piControl
(30 yrs) | Historical
(1980–2023) |
| Modified simulations in this revised manuscript | 1950control (65 yrs)
Historical (1950–2014)                        | 1950control (22 yrs)
Historical (1950–2014) | piControl
(30 yrs) | Historical
(1980–2023) |

We have also applied bootstrapping to all the EERIE simulations. Among them, only the ICON 1950control simulation failed to produce a convergent fit across the 1,000 bootstrapped resamples, likely due to variability in the underlying autocorrelation structure and the relatively shorter simulation length. Consequently, we have added the standard deviation of the e-folding timescale ( $\tau$ ) to the revised Figure 2 in the revised manuscript for all simulations except ICON 1950control (Fig. R2-1 below).

R2-1. Distribution of  $\tau$  (days) in CMIP6, EERIE coupled, and EERIE atmosphere-only (AMIP) simulations. CMIP6 and EERIE AMIP are both historical simulations, with a fixed period indicated in the x-axis labels, and the EERIE coupled simulations cover varied periods as indicated in Table 1. ERA5 is analyzed for two time periods for reference. CMIP6 results from 31 experiments are presented in violin plot, in which the width indicates the density of the data points, the

thin gray vertical box in the middle shows the  $25^{th}$  – $75^{th}$  quantiles, and the white dot presents the median. For the rest, error bars are added wherever applicable to show the  $\pm 1$  standard deviation of  $\tau$  from the 1,000 bootstrap resampling.

2. Regarding the SST sensitivity experiments: these simulations require much clearer justification. At the moment, there is very little literature which would suggest that mesoscale SST gradients should affect the SAM persistence. There are some possible physical arguments, but they are not given here. One might argue that the fact that they do appear to influence the timescale in some small ensembles using one model is justification enough, but without a solid hypothesis to test, there is no definitive answer about why the change in timescale appears. It is entirely possible that it is by chance (no estimate of sampling uncertainty is provided). An incomplete argument discusses the role of surface friction, but it requires more explicit discussion of possible mechanisms. Some additional analysis on why persistence changes grounded in possible physical mechanisms would drastically strengthen the paper. It also needs to be much clearer why two different types of mesoscale features are included. My understanding of the feedback literature is that there is no reason to expect different results from the two, and they provide basically identical results.

As suggested by the Reviewer in a point below (main comment #5), we have added a paragraph in the revised version to better contextualize our work in the existing literature, specifically regarding the processes that control SAM persistence. After this paragraph, we have developed the justification for the sensitivity experiments (see (\*) for major comment #5).

3. Regarding the consideration of friction: The literature has established methods for estimating surface friction using the model output available to the authors, but they are not followed here. Instead, the authors estimate the frictional contributions in a way which is difficult to connect to established theory of SAM persistence and the feedback parameter they calculate, and in a way that also cannot be interpreted easily across model resolutions. Its physical units are not transformed to be consistent with the momentum budget (their interpretive framework). Much more work needs to be done regarding friction if it is to be used to interpret these simulations.

To estimate the friction, we have applied two methods recommended by the reviewer in the specific comments (#16-19 below): the "faux-integration" of  $d(\rho^{-1}\tau)/dz$  (Vallis 2006, Eq. 2.270) and the residual of the momentum budget. Both approaches yielded physically consistent units (N/m²/kg\*m³/m=m/s²) and produced the correct sign within our momentum-budget framework (Simpson et al., 2012). We thank the reviewer for the suggestions. As a result, the outcomes are easier to interpret, as they follow the same conceptual framework as that employed for the eddy feedback strength. We have also extended the discussion and interpretation of the frictional impact and made connections to existing literature. For more details, please see our responses below to the specific comments #16-19.

- 4. Without significant improvements in its main areas of new contributions, the work only marginally advances knowledge in the area of SAM persistence.
  - With substantial revision and extended analyses, we consider that we have addressed the main criticisms of the reviewer and provided substantial advances on the potential role of high resolution (specifically of oceanic processes) on the simulated SAM persistence.
- 5. Finally, the work would be much stronger if it was better contextualized in the SAM feedback literature. It follows the SAM model bias literature reasonably well. Specifically, it should consider a few key areas which have important bearing on its results: 1) the evidence that the "feedback" which appears in austral summer (the focus of this work) is likely not due to eddy-mean flow interaction but nonstationary stratospheric variability (e.g., Byrne et al. 2016). 2) the understanding of SAM feedback mechanisms [barotropic (e.g. Lorenz and Hartmann 2001, Chen et al. 2008) vs baroclinic (e.g., Robinson 2000) vs diabatic (Xia and Chang 2014, Smith et al. 2024)] and how those feedback mechanisms might explain the role of surface friction and SST gradients. 3) The evidence for propagation of the SAM and for the stronger connection between SAM propagation biases and persistence biases than for eddy-feedback biases and persistence biases (e.g., Lubis and Hassanzadeh 2023). 4) the importance of the SAM timescale for climate predictability (e.g., Simpson and Polvani 2016, Ma et al. 2017, Hassanzadeh and Kuang 2019).

The main topic of our study is the ability of models to reproduce the observed SAM persistence and to determine if a better representation of meso-scale oceanic processes reduces the biases seen in lower resolution models. The introduction of our submitted manuscript was thus mainly devoted to those 'model-related' elements. However, we agree with the Reviewer that a longer discussion of the literature devoted to SAM persistence in general and specifically of the processes at the origin of this persistence would put our results in a broader context and justify more explicitly the choice of some of our analyses.

Consequently, we have added in the revised version a paragraph summarizing the main SAM feedback mechanisms. We have discussed how mesoscale oceanic features could influence the SAM persistence (point first raised in another major comment above, #2). We have also added a sentence justifying the importance of the SAM persistence for predictability. This paragraph is still relatively short considering the extensive literature on the subject. Describing all the mechanisms at play and discussing the uncertainties of those mechanisms would require a lengthy introduction considering the goal of the paper, but we come back to the relevant points in other sections of the manuscript, in particular in the conclusion when discussing some of the limitations of our study.

Paragraph to replace the lines 55-58 of the submitted version where we were defining SAM persistence:

**Lines 55-75:**

(\*) "A key characteristic of SAM is its temporal persistence, referring to how long a given phase of the SAM (positive, negative or neutral) tends to last before transitioning. This long persistence is important as it provides a source of predictability at a timescale longer than the one associated with synoptic variability (e.g., Robinson 2000; Lorenz and Hartmann 2001, Simpson and Polvani 2016). SAM persistence is often measured as the decorrelation timescale (e-folding timescale) which indicates the average duration over which the SAM index remains strongly correlated with its past values. A standard explanation attributes the extended SAM persistence to the reinforcement of westerly flow anomalies by atmospheric eddy momentum fluxes, which are generated by changes in the mean flow and act to counteract dissipation from surface friction. Several mechanisms can be at the origin of this eddy-mean flow feedback that reinforces the shifted jet, including barotropic processes related to anomalous wave propagation and breaking and baroclinic processes associated with eddy generation and enhanced baroclinicity in the lower troposphere in response to shift in the westerly flow (e.g., Robinson 2000, Lorenz and Hartmann 2001, Zurita-Gotor et al. 2014, Hassanzadeh and Kuang, 2019). The westerly flow anomalies also induce changes in the diabatic heating and cooling due to latent heat release and cloud radiative effect that modify the temperature gradients, affecting SAM persistence (Xia and Chang 2014, Smith et al. 2024, Vishny et al. 2024). In addition to this eddy-mean flow feedback, SAM persistence can have an origin from the stratosphere, which introduce some non-stationary forcing to SAM. The main influence is likely in late spring and summer at the time of the seasonal breakdown of the stratospheric vortex (Simpson et al. 2011, Byrne et al. 2016, Byrne et al. 2017, Saggioro and Shepherd 2019). Furthermore, interactions between a stationary mode and a propagating mode of the zonal variability could also affect SAM persistence (Lubis and Hassanzadeh 2021, Sheshadri and Plumb 2017, Smith et al. 2024)."

**Lines 116-123:**

"Mesoscale oceanic features strongly impact the surface heat fluxes and surface stress in the Southern Ocean -a hotspot of mesoscale activity (Frenger et al., 2013; Bishop et al. 2017). This can influence on SAM persistence as the surface heat fluxes modify the atmospheric temperature gradients, the boundary layer structure and thus diabatic heating of the atmospheric column as well as the low-level baroclinicity, which have both a demonstrated impact on SAM persistence (Xia and Chang 2014, Smith et al. 2024 Robinson 2000; Zurita-Gotor et al. 2014). Furthermore, surface stress also plays a role as it tends to damp the westerly winds but also to enhance baroclinicity and the baroclinic feedback (Robinson 2000, Zurita-Gotor 2014, Vishny et al. 2024)."

Other adjustments have been made to the introduction to avoid repetitions and make space for the new paragraphs.

**Specific Comments**

 Line 29: Assertion "eddy feedback is a better indicator" needs more justification and/or more clarity (better in what way? In what circumstances?)

This statement referred to our results from the idealized AMIP sensitivity experiments, for which the eddy feedback shows a much stronger positive correlation with the summertime  $\tau$  than the jet latitude (a clear context was already provided in the conclusion of the original manuscript). However, according to the reviewer's comments (e.g., specific comments #21, #31), we realized this statement may be misleading. We have modified the sentence to

"In these AMIP experiments, the atmospheric eddy feedback strength, combined with the damping timescale estimated via friction, correlates more strongly with  $\tau$  than  $\lambda 0$ . We speculate that the well-capture jet position (biases

Figure R2-2. Example of the SAM decorrelation timescale and eddy feedback strength calculation based on ERA5: (a) The first EOF pattern of zonal-mean geopotential anomalies; (b) The corresponding first PC time series (SAM index); (c) Autocorrelation function (ACF) of the SAM index shown for a selected day of the year (black dashed) and an exponential fit (yellow). The e-folding timescale is denoted as  $\tau$ . The ACF is repeated 1,000 times (gray) with the bootstrap sampling with replacement. (d) Same as (a) but based on vertically averaged zonal wind anomalies. (e) Lagged regression of different momentum budget terms in Eq. 3 onto the SAM index. (f) Eddy feedback strength b for positive lags 6–17 days.

12. Line 265: Because many of your models have different resolutions (particularly CMIP5 vs EERIE, you mention regridding CMIP5 to the same grid, but not EERIE), I would highly suggest following Menzel et al. (2019) or Barnes and Polvani (2015) and doing quadratic interpolation around the jet maximum to define the jet latitude. This will alleviate some of the degeneracy (models with identical jet latitudes) in Figure 3 and is consistent with the literature.

As suggested by the reviewer, we have performed a quadratic interpolation with the model output on its native grid around the jet latitude maximum to define jet latitude. While we did not observe any major difference between our previous and new results (Fig. R2-3), we have still updated our results using the new method in the revised version and added some clarifications in sec. 3.2:

L276-280: "The westerly jet is diagnosed following Menzel et al. (2019) and Barnes and Polvani (2015). We apply a quadratic fit method on the monthly mean zonally averaged 850-hPa zonal wind at the latitude where the maximum value is found between 75°S and 10°S and the four adjacent latitudes of the model. The latitude where the maximum value of the quadratic fit is found defines the position of the tropospheric westerly jet."

Figure R2-3. Jet latitude identified using two different methods, one of which is as shown in the initial version of the manuscript (y-axis) and the other follows Menzel et al. (2019) and Barnes and Polvani (2015) (x-axis).

13. Line 273: The switch from geopotential height for defining the timescale to zonal wind for defining the feedback is not without caveats. The assumption here is that the wind relevant for the SAM (and its feedbacks) is the geostrophic wind. Recently, however, Smith et al. (2024) demonstrate that SAM has significant eddy-feedbacks from the ageostrophic momentum fluxes which are leading-order in DJF in MERRA2. Vishny et al. (2024) also find important contributions to persistence from the ageostrophically-driven mean meridional circulation in idealized simulations. Thus, the imputation that models whose decay timescale is based on geopotential height will be consistent with the feedback from full (geostrophic+ageostrophic) zonal wind is probable, but not guaranteed. I think there is enough literature supporting the use of both (geopotential height and zonal wind) methods that it is not reasonable to redo the ACF calculations using zonal wind, but I do think it is worth acknowledging the geostrophic assumption and its limitations.

As suggested, a discussion of the limitations of the geostrophic assumption has been added in the revised version:

L290-294: "This shift from a definition of the SAM persistence timescale using geopotential height to the zonal wind for the estimation of the eddy-mean flow feedback is based on the standard assumption that geostrophic equilibrium provides a good approximation of the relevant variables. However, ageostrophic terms can also contribute to SAM persistence, introducing limitations to this hypothesis (Vishny et al. 2024; Smith et al. 2024)."

14. Line 278: I think the choice of three levels by Simpson et al. (2013b) was not intended to be the ideal, rather it was the best available information at the time (CMIP3). The vertical structure of SAM can

be quite nuanced despite its barotropic nature (Wall et al. 2022, Sheshadri et al. 2018), and a significant fraction of the eddy momentum flux necessary for the feedback exists above 250 hPa (Nie et al. 2014, Sheshadri et al. 2018). I suspect much more than those three levels are available for CMIP6, and there inclusion would strengthen this analysis.

We agree that including additional vertical levels would enhance the analysis. However, this choice was limited by computational constraints associated with the high-resolution EERIE simulations. As this is an international project that must balance the requirements of multiple teams within limited storage capacity, the 6-hourly data availability is currently restricted to only three vertical levels. To maintain consistency and enable comparison, we intentionally requested the same three levels used by Simpson et al. (2013b), allowing for a direct check against their results.

15. Line 300: The assumption of the Simpson framework is that the PCs are uncorrelated. Sheshadri and Plumb (2017), Lubis and Hassanzadeh (2020), Lubis and Hassanzadeh (2023), and Smith et al. (2024) have all shown this is not the case. Specifically, Sheshadri and Plumb (2017), Lubis and Hassanzadeh (2020), and Lubis and Hassanzadeh (2023) have shown that the coupling between EOF1 and EOF2 influences the SAM persistence timescale and the estimation of the eddy feedback parameter, and that the SAM timescale in CMIP6 models shows a strong dependence on the strength of the coupling between EOF1 and EOF2 (as measured by SAM's propagation period, see Lubis and Hassanzadeh 2023, Figure 7). Without examination of the coupling between modes across models, the spread in eddy-feedback parameters is difficult to interpret.

To address this comment, we have included the following paragraph in the revised version.

L326-336: "The approach followed here assumes that analyzing only the first PC is a good approximation to study SAM persistence. However, although the PCs are uncorrelated by construction on short timescale, this is not the case at longer lags and the coupling between the first two components influences SAM persistence (Sheshadri and Plumb 2017, Lubis and Hassanzadeh 2021, and Lubis and Hassanzadeh 2023). Analyzing only the first PC brings thus clear limitations in our analysis of the model spread in simulated SAM persistence. Furthermore, positive regression coefficients could be caused by non-stationarity of the series and in particular by interaction with the stratosphere and not just by eddy mean flow interactions. This introduces biases in the estimate of eddy feedback, particularly in late spring and summer (Byrne et al. 2016, Byrne et al. 2017), although this does not necessarily prevent using the regression method (Ma et al. 2017). The methodology is thus imperfect, but it provides an interpretative framework for the difference between the simulations and allows a comparison with earlier studies."

16. Line 310-325: I have two concerns involving the friction term. First, more could be done to properly estimate it and, second, utilize it in the interpretation of the results. I will begin with its estimation. Given  $\tau$  as the surface stress, one can estimate the resulting torque as  $d(\rho^{-1}\tau)/dz$  (see Vallis 2006, eq. 2.270),  $\rho$  being density. If you only have  $\tau$  at the surface, because you are vertically-integrating it, you can simply use the surface value and divide by the depth (in meters) of the atmospheric

column, as the turbulent stress is likely zero at the model top. This faux-integration also yields a net negative sign (since the stress decreases with height), and should be of the right units (N/m²/kg\*m³/m=m/s²) and the correct sign. This approach should still be approximately valid in the case that the "surface" stress is actually the output turbulent stress from the boundary layer scheme for the full boundary layer.

We have followed the reviewer's suggestions (#16 and #17) to improve estimation of the friction term (L340-350):

- (1) Since the only EERIE model output available is the turbulent surface stress, we performed the "faux-integration" by assuming (1) zero turbulent surface stress at the top of the atmospheric column, (b) fixed air density of 1.204 kg/m³ (at 15°C, 70% relative humidity and 100 kPa) and an atmosphere column depth of H=8,464 meters. As this modification introduces the multiplication of the estimate used in the submitted version by a constant factor, this would not change our cross-simulation comparison shown in the submitted manuscript but provides the friction estimation with physically correct unit and sign, allowing the results to be interpreted more easily.
- (2) To validate the estimation in (1), we have also computed the residual of the momentum budget of Eq. (3) as an alternative estimation.

Due to space constraints, the results of (1) are presented in the main manuscript, while the comparison between (1) and (2) is provided in the supplementary material (Fig. R2-4a, b). Our results show that the two methods differ in their absolute magnitudes (not surprising given the simplified assumptions), but the inter-simulation comparisons are consistent between the two methods. The negative linear regression between the estimated friction term and  $\tau$  is also not strongly affected by the employed methods.

Figure R2-4. (a) Scatter plot of  $\tau$  and the frictional impact estimated as the residual of Eq.3 in the main article (black and red markers are for ObsSST and NoEddies experiments, respectively; star markers indicate the 9-km simulations and the rests are the 28-km runs; yellow circle represents ERA5). (b) Scatter plot of  $\tau$  and the eddy feedback strength, b. (d) Scatter plots of  $\tau$  and the combined effects of friction (expressed as Rayleigh damping timescale,  $t_f$ ) and eddy feedback strength b, measured as  $t_f/(1-b\cdot t_f)$  following Lorenz and Hartmann (2001). In all subplots, the dotted gray line represents the linear regression fit, and the correlation coefficient and p value are shown in the top-right corner.

17. However, the friction in Lorenz and Hartmann (2001; and in other studies building on this framework) is generally parameterized as Rayleigh drag with a constant damping timescale. LH2001 explain in their Appendix A how to estimate it from timeseries of m and z. Since both of these fields are used in this analysis, it should be possible to estimate a friction via Rayleigh drag. This has two key benefits: 1) it can be used to validate the friction estimated from the stress, and triple checked against the residual of the momentum budget, evaluated from your equation (3), which should also be possible. In my experience, the residual usually matches the Rayleigh drag

quite well. The second benefit is that it is useful for the interpretation of the feedback parameter, which I discuss more later.

Lorenz and Hartmann (2001) quantified the eddy feedback and the frictional term using spectral analysis and cross covariance. Here, we follow a different approach to estimate the feedback parameter, following Simpson et al. (2013b) (see for instance their appendix for a justification). To additionally compute the frictional term following LH2001 would require too much extra space to explain the different methodology and introduce more complexity in our discussion, and so we have decided not to apply this approach. By contrast, we have computed the friction term as a residual of the momentum budget as suggested.

18. A final issue with the estimation of the friction (no matter which method, preferably at least 2 of the 3) is that its projection value is proportional to the square root of the number of latitudes, and thus its magnitude should not be compared directly with simulations with different horizontal resolution. This is true for all the budget terms, but the feedback parameter is resolution-independent because it involves the ratio of two budget terms. To understand why, consider a simplified version of your equation (2) where  $\mathbf{W} = \mathbf{I}$  (the identity). If  $\mathbf{e}$  is a (square-) integrable function  $f(\lambda)$ , sampled on an equally spaced grid (reasonable for GCM output), its Euclidean norm will be proportional to the square-root of the integral of  $[f(\lambda)]^2$  over latitude ( $\lambda$ ), divided by the grid spacing (since we multiplied by the grid spacing to convert the sum into an integral). The integral should converge to the same value regardless of the resolution for most smoothly-varying, well-resolved f (again reasonable at even coarse GCM resolutions). However, the inverse of the grid spacing is proportional to the number of latitudes N (if the grid is evenly spaced). Thus the norm of **e** is proportional to  $\sqrt{N}$ . The multiplication of Xe is proportional to N (not the square root), by the same logic (because e is an orthogonal basis, the only component of **X** that survives the integration is proportional to **e**, and the product is proportional to **ee**). However, **Xe** has no square root, and thus **Xe**/ $\sqrt{ee}$  is proportional to  $\sqrt{N}$ . See a small example which should generalize well as the attached image. Note that including a non-identity weighting matrix **W**≠**I** does not change this, it simply adds another term into the integration. One could divide by  $\sqrt{N}$  to alleviate this, or use integrals in the top and bottom instead. Or, one could divide the friction by the zonal wind projection as done for the feedback parameter. At that point, you may as well compute the damping timescale following the literature (LH2001, Appendix A).

We thank the Reviewer for pointing this out. We mentioned that all the analysis were performed on the EERIE model outputs interpolated onto a  $0.25^{\circ} \times 0.25^{\circ}$  grid (Line 133-134 in the original manuscript) to facilitate direct comparison across experiments. This has now been made more explicit in the revised manuscript (L144-146). As a result, differences in the models' native grid resolutions do not introduce relative bias in this comparison. Nonetheless, we agree that this is an

important point to emphasize. We have therefore included a note in the revised manuscript (L351-353) highlighting the dependence of such a calculation on data resolution.

While we have chosen to retain the projected friction term in Fig. 4, we have also calculated a damping timescale (*tr*) using the residual term of Eq. (3), equivalent to the one corresponding to a Rayleigh drag, i.e.,

$$[F]s = -[u]s/t_{f}$$

Instead of following the spectrum approach as in LH2001, here we follow the same framework for the eddy feedback strength estimation of Simpson et al. (2013b) by performing the lagged linear regression of [F]s and [u]s onto the PC(t). Finally,  $t_f$  can be estimated by taking the ratio between the regressed [u]s (in unit of m/s) and the regressed [F]s (unit of m/  $s^2$ ) averaged over the lag days of 7-14 days. Our resultant damping timescale is 8.6 days for ERA5, very close to the 8.9 days estimated in LH2001 (L472-476).

19. The second friction-related issue is with the interpretation of the feedback. Following LH2001, the eddy feedback parameter (b) lengthens the effective timescale for the SAM by tf/(1-b\*tf), where tf is the frictional timescale. Thus, both the eddy feedback and the frictional timescale can effect SAM's persistence, and if models have differing frictional timescales, it could also explain differences in their persistence. In theory, one could see if this effective timescale tf/(1-b\*tf) followed the autocorrelation timescale more closely (I suspect it would), but the model bias literature (Gerber et al. 2008, Kidston and Gerber 2010) generally does not follow this convention, so I don't think this is strictly necessary. However, it may give a better interpretive framework for the friction to plot the frictional timescale (rather than the projection) and use this LH2001 relation to explain how the frictional timescale interacts with the eddy-feedback parameter to determine the total timescale.

As indicated above, we have computed a similar estimation of the damping timescale (tf) to extend our interpretation to the existing literature. This allows us to examine the quantity of  $tf / (1 - b \cdot tf)$  (Fig. R2-4d). In our results, this quantity indeed shows a stronger positive correlation with SAM persistence  $\tau$  with a larger correlation coefficient and lower p-value compared to when using b as the sole predictor variable. This quantity therefore more effectively describes the joint impact of friction and large eddy feedback on SAM persistence. However, like b and friction, this quantity does not provide a clear explanation for the subtle sensitivity of  $\tau$  to different SST boundary conditions and the model resolution. Nevertheless, we have provided this additional diagnostic in the supplementary material.

20. Figure 2 (caption): Please describe the violin plot in more detail; I don't believe they are common enough to assume they can be interpreted properly without explanation.

We have modified the Figure 2 caption to

"Distribution of  $\tau$  (days) in CMIP6, EERIE coupled, and EERIE atmosphere-only (AMIP) simulations. CMIP6 and EERIE AMIP are both historical simulations, with a fixed period indicated in the x-axis labels, and the EERIE coupled simulations cover varied periods as indicated in Table 1. ERA5 is analyzed for two time periods. **CMIP6 results from 31 experiments are presented in violin plot, in which the width indicates the density of the data points, the thin gray vertical box in the middle shows the 25th –75th quantiles, and the white dot presents the median.** For the rest, error bars are added wherever applicable to show the  $\pm 1$  standard deviation of  $\tau$  from the 1,000 bootstrap resampling."

21. Lines 396-398: I think this point on the interpretation of the IFS-AMIP experiments requires more discussion and computation. These are an IC ensemble from the same GCM with the same boundary conditions, and thus represent internal variability of the same mean climate in a way that isn't the case for comparisons across the CMIP models. For example, you could likely run 5 more IC ensembles, and you might get a completely different pattern between jet latitude and e-folding timescale. But I don't think that somehow contradicts the expectation that the two should be positively correlated due to the stronger wave reflection (and weaker feedback) of more poleward jets (Barnes and Hartmann 2010, Lorenz 2023). Despite this, according to the convergence estimates of Gerber et al. (2008), the 40 years of AMIP simulations should be enough to constrain the decorrelation timescale within a day, and 4 of the ensemble members are within one day of their mean. This is where I think bootstrapped estimates of the sampling uncertainty could help resolve this question of whether sampling uncertainty can explain the lack of relationship, or whether this is indeed a breakdown of the expected theory.

We agree with the Reviewer that the IC ensemble is different from CMIP models as the boundary conditions are the same for all the IC members while ensemble of coupled models can have very different SST patterns, for instance. This has been made more explicit in the revised version by mentioning 'internal atmospheric variability' instead of just 'internal variability' (L422). We agree with the Reviewer and we also "…don't think that somehow contradicts the expectation that the two should be positively correlated due to the stronger wave reflection (and weaker feedback) of more poleward jets (Barnes and Hartmann 2010, Lorenz 2023)." We agreed that the documented relationship between biases in  $\tau$  and  $\lambda$ 0 in the literature is based on strong physical arguments. However, the sentences referenced here (original L396-398) are purely descriptive of the shown results that there is no clear positive correlation between  $\tau$  and  $\lambda_0$  among IFS-AMIP simulations. The hypothesis we put forward is that, when the position of the jet is well captured as in all IFS-AMIP experiments (due to the constrained SST boundary conditions or not), the difference in jet position is too small between these experiments (compared for instance with CMIP models) to explain the difference in decorrelation timescale and other factors dominates, but this is only a speculation at

this stage. We have also examined the uncertainty via bootstrapping, but it doesn't help to explain the lack of such a relationship here. Finally, as noted in our reply to a comment below (#31), we have modified the conclusion to make it clear that that the AMIP results are not interpreted as evidence that jet latitude is irrelevant.

22. Figure 3: When uncertainty exists in both the dependent and independent variables of a regression, it may be more appropriate to use a different type of regression than least-squares, especially if the uncertainties are correlated (see Pendergrass and Kao 2022, and York 2004 for an alternative).

We agree that a more sophisticated regression method could be more precise, but we have chosen here the standard least-squares for simplicity as in some previous studies.

23. Line 415: Sample size of one, not enough evidence to support conclusion (bootstrapping would help)

We have added the standard deviation of  $\tau$  for all AMIP simulations using bootstrap resampling. The decrease in  $\tau$  with higher resolution is not particularly robust on the annual-mean scale, as the estimated  $\tau$  in the 9-km simulation falls within (though toward the lower end of) the range covered by the 28-km simulations (Fig. R2-1a). However, during the NDJ season, the reduction in  $\tau$  for the 9-km simulation is more pronounced, with its estimated range extended outside those of the 28-km simulations (Fig. R2-1b).

24. Lines 425-430: Some connection to existing feedback mechanisms would be appropriate here

We have added some connections to the existing literature in the revised manuscript. See also our responses to specific comments #16-#19.

25. Lines 441-449: See previous comments regarding friction

Addressed in responses to specific comments #16-#19.

26. Line 462: Is the convection parameterization turned off at 9km? Stronger latent heating in the 9km run could create a stronger negative diabatic feedback (Xia and Chang 2014), decreasing the persistence

The convection parameterization is still active at 9 km, as at 28 km resolution, consistently with the convection parameterization settings applied in ECMWF operational forecasts (added in L176).

27. Figure 4: Panels (f), (g) and (h) should be greatly simplified, maybe down to one panel (or even a table), showing the simulation on one axis and the value of the x axis on the other. The decay timescales are identical, and two points is not enough to infer any relationship, so the current

scatter plots visually complicate the comparison between simulations. Readers will understand why they do not follow (b), (c), and (d), no need to artificially fit that pattern.

We have removed the original panels (f), (g) and (h) in Fig. 4, showing the 9-km results on the same scatter plots as the 28-km members (shown here as Fig. R2-5).

Figure R2-5. (a) SAM decorrelation timescale ( $\tau$ ) as a function of month for IFS-AMIP 28km simulations (dashed for each ensemble member and solid for the ensemble means; black for ObsSST and red for NoEddies experiments) and ERA5 (yellow). (b) Similar to (a) but for 9 km experiments (shades for the  $\pm 1$  standard deviation of  $\tau$  from the 1,000 bootstrap resampling). (c) Scatter plot of  $\tau$  (days; y-axis) and westerly jet latitude (x-axis; filled-color markers for 28 km; hollow stars for 9 km simulations). (d)–(e) Similar to (c) but with x-axis variable replaced with the eddy feedback strength and frictional impact, respectively. In (b)–(d), the gray dotted line represents the linear regression fit, and the correlation coefficient and p-value are indicated in the top-right corner.

28. Line 492: For the 10 year, coupled EERIE simulations, I'm not convinced this is long enough to really reduce the sampling uncertainty, which converges very slowly (see Gerber et al. 2008). Bootstrapped measures would help alleviate this concern; without such attempts, it is hard to interpret the difference between the EERIE simulations and the longer CMIP6 simulations (and the longer IFS-AMIP simulations for that matter).

As our responses to major comment #1, we have replaced the previously shown EERIE coupled simulations with much-extended ones (two 65-year simulations with IFS-FESOM2, one 30-year

control run with HadGEM3, and one 22-year control and one 65-year historical runs with ICON) and apply bootstrapping resampling to strengthen our analysis.

**29. Line 540: please clarify: better indicator of what?**

Please see our response to specific comment #1.

30. Line 541: "more statistically significantly" If I recall, while the p-value was small, the result was not significant. I think significance is too binary for this language. I would stick with language which discusses what a small p-value means (the relationship is unlikely to be due to random chance).

The lower the p-value indicates that we have higher confidence to conclude that there is a significant linear relationship between the two variables (i.e., higher confidence to reject the null hypothesis of no correlation). We think p-value is standard and does not need to be further explained. However, we have rephrased the lines:

"Compared to  $\lambda_0$ , the metric eddy feedback strength b shows a much stronger correlation with SAM persistence  $\tau$ , with a higher correlation coefficient of 0.52 and a lower p-value of 0.08 (Fig. 4d), suggesting it may be a more informative indicator of SAM persistence in this configuration. Meanwhile, the surface friction and  $\tau$  exhibit a negative correlation (Fig. 4e) with a moderate correlation coefficient of -0.48 and p-value of 0.11." (L462-466)

31. Line 522: I'm not convinced the path forward is that promising from these results. A higher resolution atmosphere helps. That is good. But it does not seem to benefit from being coupled (bias improves in AMIP) and it does not seem to benefit from mesoscale ocean features (smoothed SST runs have lower bias). Improvements in jet latitude at these resolutions do not seem to help either. However, the climate community will want to run coupled models for the estimation of climate variability and sensitivity for the foreseeable future. If other models behave like IFS (a big assumption), it is likely models will be stuck with some irreducible bias in SAM timescale. Perhaps I am too pessimistic. If so, please help me understand what other path these results suggest.

Proposing a clear path forward based on these results is inherently challenging and subject to considerable uncertainty. As the reviewer rightly points out, our AMIP simulations suggest that model performance in representing SAM persistence does not clearly benefit from two-way ocean-atmosphere coupling or from the explicit inclusion of ocean mesoscale features.

Our hypothesis is that while coupled models offer a more physically consistent representation of the climate system, they also tend to introduce SST biases—potentially due to under-tuning in high-resolution configurations or imbalances in the coupling process. As demonstrated by our AMIP experiments, which use prescribed SST boundaries, these SST biases have a more pronounced

effect on SAM characteristics than the explicit representation of air—sea coupling. In fact, previous studies have shown that eddy-permitting models can exhibit larger SST biases than either coarser models with parameterized eddy fluxes or fully eddy-rich models (e.g., Storkey et al. 2025). Although our AMIP SST-varied experiments do not identify a robust, direct impact of ocean mesoscale eddies on SAM persistence, their localized influence in the atmospheric boundary layer and critical roles in the Southern Ocean climate system are well documented. We conclude that while ocean eddies' local impact on the atmospheric boundary layer is well established, their direct influence in modulating large-scale modes such as the SAM appears limited under our AMIP setup without air-sea coupling.

A key takeaway from the above is that reducing SST biases remains essential for advancing the representation of SAM and Southern Hemisphere climate variability. It is likely that only with more accurate SST fields can the climate modeling community properly assess the role of ocean mesoscale processes.

Another point we would like to clarify is that overall, the high-resolution EERIE models—both coupled and uncoupled—show improvement in certain aspects of SAM variability, indicating that increased resolution can offer benefits. Specifically, the reduction in  $\tau$  bias relative to CMIP6 models is accompanied by improved representation of jet latitude. Consistent with the existing literature, a correlation between  $\tau$  and jet latitude is also found in the EERIE simulations (as shown in our Fig. 3). Although this relationship is virtually absent in the atmosphere-only AMIP experiments—characterized by a negligible slope and a large p-value—this should not be interpreted as evidence that jet latitude is irrelevant. Rather, it highlights that when the jet is already well captured (with <1° bias) and SSTs are prescribed, other second-order processes may come into play to affect  $\tau$ .

In light of the reviewer's comments, we have revised the Discussion and Conclusions section to more clearly articulate these implications and highlight the broader relevance of our findings.

**Technical Corrections**

Lines 73, 90, 240: Simpson 2013 referenced without a/b

Corrected, Thanks.

Lines 376-379: This sentence "However, it is also possible... more critical" could benefit from more clarity, including maybe breaking into smaller sentences.

The sentence has been shortened to "However, it is also possible that the improvements typically attributed to higher resolution on the performance of large-scale SAM variability and the mean jet have reached a plateau at the grid sizes used in current GCMs (e.g., CMIP6)." (L404-406)

Lines 511-514: Rephrasing (and separating into smaller sentences) would improve clarity here

We have rephrased the sentence to "While Sen Gupta and England (2006) showed that air-sea coupling is critical for modulating the SAM—albeit focusing on interseasonal timescales, which are longer than the intraseasonal scale investigated here—our results suggest that atmosphere-ocean coupling plays a secondary role. Instead, SST biases introduced by the coupling—an ongoing challenge in coupled GCMs (Zhang et al., 2023)—appear to be more influential." (L525-529)

**References**

Barnes, E. A., & Hartmann, D. L. (2010). Testing a theory for the effect of latitude on the persistence of eddy-driven jets using CMIP3 simulations. Geophysical Research Letters, 37(15). https://doi.org/10.1029/2010GL044144

Barnes, E. A., & Polvani, L. M. (2015). CMIP5 Projections of Arctic Amplification, of the North American/North Atlantic Circulation, and of Their Relationship. Journal of Climate, 28(13), 5254–5271. https://doi.org/10.1175/JCLI-D-14-00589.1

Byrne, N. J., Shepherd, T. G., Woollings, T., & Plumb, R. A. (2016). Annular modes and apparent eddy feedbacks in the Southern Hemisphere. Geophysical Research Letters, 43(8), 3897–3902. https://doi.org/10.1002/2016GL068851

Byrne, N. J., Shepherd, T. G., Woollings, T., & Plumb, R. A. (2017). Nonstationarity in Southern Hemisphere Climate Variability Associated with the Seasonal Breakdown of the Stratospheric Polar Vortex. Journal of Climate, 30(18), 7125–7139. https://doi.org/10.1175/JCLI-D-17-0097.1

Chen, G., Lu, J., & Frierson, D. M. W. (2008). Phase Speed Spectra and the Latitude of Surface Westerlies: Interannual Variability and Global Warming Trend. Journal of Climate, 21(22), 5942–5959. https://doi.org/10.1175/2008JCLI2306.1

Gerber, E. P., Voronin, S., & Polvani, L. M. (2008). Testing the Annular Mode Autocorrelation Time Scale in Simple Atmospheric General Circulation Models. Monthly Weather Review, 136(4), 1523–1536. https://doi.org/10.1175/2007MWR2211.1

Hassanzadeh, P., & Kuang, Z. (2019). Quantifying the Annular Mode Dynamics in an Idealized Atmosphere. Journal of the Atmospheric Sciences, 76(4), 1107–1124. <a href="https://doi.org/10.1175/jas-d-18-0268.1">https://doi.org/10.1175/jas-d-18-0268.1</a>

Kidston, J., & Gerber, E. P. (2010). Intermodel variability of the poleward shift of the austral jet stream in the CMIP3 integrations linked to biases in 20th century climatology. Geophysical Research Letters, 37(9). https://doi.org/10.1029/2010GL042873

Lorenz, D. J. (2023). A Simple Mechanistic Model of Wave–Mean Flow Feedbacks, Poleward Jet Shifts, and the Annular Mode. Journal of the Atmospheric Sciences, 80(2), 549–568. https://doi.org/10.1175/JAS-D-22-0056.1

Lorenz, D. J., & Hartmann, D. L. (2001). Eddy–Zonal Flow Feedback in the Southern Hemisphere. Journal of the Atmospheric Sciences, 58(21), 3312–3327. <a href="https://doi.org/10.1175/1520-0469(2001)058<3312:EZFFIT>2.0.CO;2">https://doi.org/10.1175/1520-0469(2001)058<3312:EZFFIT>2.0.CO;2</a>

Lubis, S. W., & Hassanzadeh, P. (2020). An Eddy–Zonal Flow Feedback Model for Propagating Annular Modes. Journal of the Atmospheric Sciences, 78(1), 249–267. https://doi.org/10.1175/JAS-D-20-0214.1

Lubis, S. W., & Hassanzadeh, P. (2023). The Intrinsic 150-Day Periodicity of the Southern Hemisphere Extratropical Large-Scale Atmospheric Circulation. AGU Advances, 4(3), e2022AV000833. https://doi.org/10.1029/2022AV000833

Ma, D., Hassanzadeh, P., & Kuang, Z. (2017). Quantifying the Eddy–Jet Feedback Strength of the Annular Mode in an Idealized GCM and Reanalysis Data. Journal of the Atmospheric Sciences, 74(2), 393–407. https://doi.org/10.1175/JAS-D-16-0157.1

Menzel, M. E., Waugh, D., & Grise, K. (2019). Disconnect Between Hadley Cell and Subtropical Jet Variability and Response to Increased CO2. Geophysical Research Letters, 46(12), 7045–7053. https://doi.org/10.1029/2019GL083345

Nie, Y., Zhang, Y., Chen, G., Yang, X.-Q., & Burrows, D. A. (2014). Quantifying barotropic and baroclinic eddy feedbacks in the persistence of the Southern Annular Mode. Geophysical Research Letters, 41(23), 8636–8644. https://doi.org/10.1002/2014GL062210

Robinson, W. A. (2000). A Baroclinic Mechanism for the Eddy Feedback on the Zonal Index. Journal of the Atmospheric Sciences, 57(3), 415–422. <a href="https://doi.org/10.1175/1520-0469(2000)057<0415:ABMFTE>2.0.CO;2">https://doi.org/10.1175/1520-0469(2000)057<0415:ABMFTE>2.0.CO;2</a>

Saggioro, E., & Shepherd, T. G. (2019). Quantifying the Timescale and Strength of Southern Hemisphere Intraseasonal Stratosphere-troposphere Coupling. Geophysical Research Letters, 46(22), 13479–13487. <a href="https://doi.org/10.1029/2019GL084763">https://doi.org/10.1029/2019GL084763</a>

Sheshadri, A., & Plumb, R. A. (2017). Propagating Annular Modes: Empirical Orthogonal Functions, Principal Oscillation Patterns, and Time Scales. Journal of the Atmospheric Sciences, 74(5), 1345–1361. https://doi.org/10.1175/JAS-D-16-0291.1

Sheshadri, A., Plumb, R. A., Lindgren, E. A., & Domeisen, D. I. V. (2018). The Vertical Structure of Annular Modes. Journal of the Atmospheric Sciences, 75(10), 3507–3519. https://doi.org/10.1175/JAS-D-17-0399.1

Simpson, I. R., Hitchcock, P., Shepherd, T. G., & Scinocca, J. F. (2011). Stratospheric variability and tropospheric annular-mode timescales. Geophysical Research Letters, 38(20). https://doi.org/10.1029/2011GL049304

Simpson, Isla R., & Polvani, L. M. (2016). Revisiting the relationship between jet position, forced response, and annular mode variability in the southern midlatitudes. Geophysical Research Letters, 43(6), 2896–2903. https://doi.org/10.1002/2016GL067989

Simpson, Isla R., Hitchcock, P., Shepherd, T. G., & Scinocca, J. F. (2013). Southern Annular Mode Dynamics in Observations and Models. Part I: The Influence of Climatological Zonal Wind Biases in a Comprehensive GCM. Journal of Climate, 26(11), 3953–3967. <a href="https://doi.org/10.1175/JCLI-D-12-00348.1">https://doi.org/10.1175/JCLI-D-12-00348.1</a>

Simpson, Isla R., Shepherd, T. G., Hitchcock, P., & Scinocca, J. F. (2013). Southern Annular Mode Dynamics in Observations and Models. Part II: Eddy Feedbacks. Journal of Climate, 26(14), 5220–5241. <a href="https://doi.org/10.1175/JCLI-D-12-00495.1">https://doi.org/10.1175/JCLI-D-12-00495.1</a>

Smith, S., Lu, J., & Staten, P. W. (2024). Diabatic Eddy Forcing Increases Persistence and Opposes Propagation of the Southern Annular Mode in MERRA-2. Journal of the Atmospheric Sciences, 81(4), 743–764. https://doi.org/10.1175/JAS-D-23-0019.1

Vallis, G. K. (2006). Atmospheric and Oceanic Fluid Dynamics (1st ed.). Cambridge University Press.

Vishny, D. N., Wall, C. J., & Lutsko, N. J. (2024). Impact of Atmospheric Cloud Radiative Effects on Annular Mode Persistence in Idealized Simulations. Geophysical Research Letters, 51(15), e2024GL109420. https://doi.org/10.1029/2024GL109420

Wall, C. J., Lutsko, N. J., & Vishny, D. N. (2022). Revisiting Cloud Radiative Heating and the Southern Annular Mode. Geophysical Research Letters, 49(19), e2022GL100463. <a href="https://doi.org/10.1029/2022GL100463">https://doi.org/10.1029/2022GL100463</a>

Xia, X., & Chang, E. K. M. (2014). Diabatic Damping of Zonal Index Variations. Journal of the Atmospheric Sciences, 71(8), 3090–3105. https://doi.org/10.1175/JAS-D-13-0292.1

York, D., Evensen, N. M., Martínez, M. L., & De Basabe Delgado, J. (2004). Unified equations for the slope, intercept, and standard errors of the best straight line. American Journal of Physics, 72(3), 367–375. https://doi.org/10.1119/1.1632486

Zurita-Gotor, P. (2014). On the Sensitivity of Zonal-Index Persistence to Friction. Journal of the Atmospheric Sciences, 71(10), 3788–3800. <a href="https://doi.org/10.1175/JAS-D-14-0067.1">https://doi.org/10.1175/JAS-D-14-0067.1</a>

Zurita-Gotor, P., Blanco-Fuentes, J., & Gerber, E. P. (2014). The Impact of Baroclinic Eddy Feedback on the Persistence of Jet Variability in the Two-Layer Model. Journal of the Atmospheric Sciences, 71(1), 410–429. https://doi.org/10.1175/JAS-D-13-0102.1

**Our Reference**

Bishop, S. P., Small, R. J., Bryan, F. O. & Tomas (2017). R. A. Scale dependence of midlatitude air–sea interaction. J. Clim. 30, 8207–8221. https://doi.org/10.1175/JCLI-D-17-0159.1.

Frenger, I. et al. (2013). Imprint of Southern Ocean eddies on winds, clouds and rainfall. Nature Geoscience 6, 608–612.

Roberts, C., Aengenheyster, M., Roberts, M. (2024b). Description of EERIE idealised atmosphere-only simulations. https://doi.org/10.5281/ZENODO.14514510

Storkey D, P. Mathiot, M. J. Bell, D. Copsey, C. Guiavarc'h, H. T. Hewitt, J. Ridley, and Ma. J. Roberts (2025). Resolution dependence of interlinked Southern Ocean biases in global coupled HadGEM3 models. Geosci. Model Dev., 18, 2725–2745, 2025. https://doi.org/10.5194/gmd-18-2725-2025

---

## Author Response (AR2)

**Response** to referees' comments on "Southern Annular Mode Persistence and Westerly Jet: A Reassessment Using High-Resolution Global Models" by Chen et al.

MS No.: egusphere-2025-666 MS type: Research article

**Reviewer 2**

The revised manuscript makes a number of important changes: 1) the analysis of surface friction is more thorough, 2) the choice of model simulations to analyze are refined, with many simulations running for longer, 3) the manuscript is better situated in the literature, 4) it conclusions follow in a more logically-consistent manner from the results, with better discussion of relevant limitations, 5) the figures include more relevant information in regards to uncertainty and less extemporaneous information,6) the narrative has a tighter focus on the SAM timescale across the different simulation types, and 7) general improvements to the writing quality. With these revisions, my previous major concerns regarding the manuscript are satisfied, and I find the paper to be generally acceptable for publication, with some minor corrections. The figures in particularly are of high quality. I also suggest a general edit of the manuscript to tighten the language, which will improve clarity.

We appreciate reviewer 2's positive feedbacks. For the remaining comments and suggestions, please see our responses below in blue.

**Scientific Comments**

Lines 241-245: What is the filter being used here? Adding this would help align with the stated goal of reproducibility. Additionally, you might add a brief summary for the motivation for the double filtering (60-day then 30-year, which technically overlap). I think this procedure is not intuitive and currently requires readers to reference Gerber et al. (2010) to understand that this is a regularization technique. I think it would help the description to make it more explicit that the 30-year filter is being applied to the 60-day filtered data (at first I thought both were derived from the original time series, although the text does not say that. Mostly I find the text a bit ambiguous). I had to read the description several times to pick up on this subtlety. I understand this is a complex technique which is generally following previous literature, and the authors are doing quite well already, but a little further revision would help.

We applied the discrete Fourier transform (DFT) on the time series and filters out signals with a frequency higher than the cutoff frequency of 60 days. The double-filtering procedure follows exactly Gerber et al. (2010). We have clarified the procedure (and their reasoning) in the revised manuscript:

"Such  $\Phi(\lambda,t)$  field is derived in two steps following Gerber et al. (2010). To avoid overfitting high-frequency noise, a 60-day low-pass filter is first applied to the detrended  $\Phi(\lambda,t)$  along the t axis to retain only seasonal-scale variability. Specifically, we apply the discrete Fourier transform to the time series and filter out components with frequencies higher than 1/60 days-1. The resulting smoothed time series is then reindexed by calendar day (d) and year (y). For each calendar day (e.g., Jan 1st, Jan 2nd, etc.), a 30-year low-pass filter is subsequently applied along the y axis to extract long-term variations."

Lines 320-321: "Assuming that at sufficiently large positive lags, the feedback component dominates the eddy forcing," -> "Assuming that the random component of the eddy forcing is uncorrelated at sufficiently large positive lags,". I personally find that the decorrelation between the stochastic forcing and the original anomalies is a clearer explanation (since truly random forcing should not be state-dependent), but disagreeing would be reasonable. Some slight rephrasing may still be worthwhile.

Thank you. We agree to change the sentence as suggested.

Line 383: The timescale is lower with 9km, but you might recognize this may not be due to the change in resolution since the error bars overlap with the 28km ones, especially for the annual case.

We have decided to keep the sentence but add a short clause about the overlapping error bars to acknowledge the uncertainty: "Refining the atmospheric resolution from 28 km to 9 km suggests a lowering of the SAM decorrelation timescale, with  $\tau$  of 8 days annually and 10 days in NDJ. However, the difference may not be robust, as the bootstrapped error bars of both resolutions overlap."

Line 402: You might add a brief statement before "A potential dependency on resolution...", referring to the previous result that EERIEE models have higher resolution and lower persistence. Just to help your audience understand why we may expect such dependence. Or you could discuss part of Gerber et al. (2008), currently in Lines 407-410, which suggests higher resolution may reduce persistence biases (but then wait to talk about the plateau effect until later).

Thank you for the good advice. We have modified the paragraph as "As EERIE results suggest that higher resolution may reduce persistence biases, we examine the model resolution of each CMIP6 simulation. However, there appears no strong or clear relationship between the model resolution and the model biases in either  $\tau$  or  $\lambda$  (the conclusion holds for both latitudinal and longitudinal resolutions and for both atmospheric and oceanic components, although only the atmospheric latitudinal resolution is expressed in Fig. 3). A potential dependency on resolution could be obscured in the CMIP6 ensemble by other compensating factors arising from different model configurations and system designs."

Line 534: I wouldn't describe these mechanisms as competing. The effect of jet latitude on SAM is generally argued to be related to the eddy feedback, with more poleward jets requiring stronger wave breaking and larger feedbacks. Of course jet latitude isn't the only factor affecting the feedback, as this work indicates. I'd search for a different word to describe their relationship.

By "two competing dominant mechanisms", we meant the eddy momentum flux convergence (as measured in our diagnostic eddy feedback) and the surface friction. We have removed that term to avoid confusion: "Indeed, we find that the metrics of atmospheric eddy-mean feedback strength, surface friction and their joint effect correlate more strongly with  $\tau$  than with  $\lambda_0$  in the AMIP configurations, highlighting the importance of these mechanisms on SAM persistence."

**Technical Corrections**

Line 41: "inferred" -> "implied"

Corrected

Lines 64-70: This is a good summary of the relevant mechanisms. I suggest restructuring the writing to improve clarity.

We have rewritten the section to: "Several mechanisms may contribute to the eddy—mean flow feedback that reinforces the shifted jet. These include barotropic processes, such as anomalous wave propagation and breaking, and baroclinic processes related to enhanced eddy generation and increased lower-tropospheric baroclinicity in response to shifts in the westerly winds (e.g., Robinson 2000, Lorenz and Hartmann 2001, Zurita-Gotor et al. 2014, Hassanzadeh and Kuang, 2019). Westerly flow anomalies also induce changes in the diabatic heating and cooling—through latent heat release and cloud radiative effects—which alter temperature gradients and, in turn, affect SAM persistence (Xia and Chang 2014, Smith et al.2024, Vishny et al. 2024)."

Line 76: "Skills" -> "Skill"

Corrected

Line 137-138: "the ocean eddies" -> "mesoscale eddies"

Changed

Line 148: "are conducted" -> "were conducted" or "are being conducted" (if ongoing)

Changed to "are being conducted"

Line 150: "similar as" -> "similar to"

Corrected

Lines 160-170: Well said.

Thanks

Line 192: "conditions with the a spatial" -> "conditions with a spatial"

Corrected

Line 195: "their relative winds-currents effects" -> "their mechanical effects"

The sentence has been rephrased to "We emphasize that such a design only allows us to test ocean eddies' direct thermodynamic impact (as reflected in SSTs) but not their mechanical influence (through the so- called wind stress feedback or relative winds-currents effects)"

Line 198: "spatially varying climatological" -> "spatially varying, climatological"

Added

Line 304: "zonal momentum tendency equation" -> "zonal momentum equation"

Changed

Line 327: "However, although the PCs are uncorrelated by construction on short timescale," -> "While the PCs are uncorrelated on short timescales (by construction),"

Modified as suggested. Thanks.

Line 426: Nice transition.

**Thanks**

Figure 4 (caption): I think the marker shapes are just individual ensemble members (only the star is discussed), but you might mention their significance (or lack thereof). I would also suggest using one legend for the whole figure since the color scheme seems to imply some of the simulations are ObsSST and some NoEddies, but I don't find this explicitly stated anywhere. Right. We have modified the caption for figure 4 to provide clearer explanation about the shown components:

"Figure 4. Analysis of the IFS-AMIP idealized experiments (black for ObsSST and red for NoEddies; yellow for ERA5 as reference): (a) SAM decorrelation timescale ( $\tau$ ) as a function of

month for 28km simulations (dashed for individual ensemble members and solid for ensemble means). (b) Similar to (a) but for 9 km experiments (shades for the  $\pm 1$  standard deviation of  $\tau$  from the 1,000 bootstrap resampling). (c) Scatter plot of  $\tau$  (days; y-axis) and westerly jet latitude (x-axis; filled-color markers for 28 km; hollow stars for 9 km simulations). (d)–(e) Similar to (c) but with x-axis variable replaced with the eddy feedback strength and frictional impact, respectively. In (b)–(d), the same marker shape indicates the same ensemble member. The gray dotted line represents the linear regression fit, and the correlation coefficient and p-value are indicated in the top-right corner."

Figure 4e: Perhaps my eyes are seeing what they want, but it seems the frictional impact is lower in the NoEddies during DJF, consistent with the lower persistence. It seems unlikely to be a robust relationship, but maybe it is worth quantifying?

While three NoEddies members do show lower frictional impact, consistent with their lower persistence, the strongest frictional impact (red square) also occurs in one of the NoEddies members. This variability makes it challenging to conclude a systematic shift between ObsSST and NoEddies. Nonetheless, we agree that this potential relationship is interesting and merits further investigation in future work.

**Lengthy Phrases**

I suggest the following phrases/sentences be made more concise. Ellipsized sentences refer to the entire sentence. The list is not exhaustive. I provide a few sample rewrites as suggestions.

We have taken the following editorial suggestions or rewritten the marked sentences to improve readability. We've also re-read the manuscript and made a few more changes to avoid lengthy phrases.

Line 26-27: "results in reduced tau" -> "reduces tau"

Changed accordingly

Line 62: "act to counteract" -> "counteract"

Changed accordingly

Lines 63-64: "Several mechanisms can be at the origin of this eddy-mean flow feedback that reinforces the shifted jet..."

Modified. See our response above.

Lines 81-84: "Many studies have found a strong dependency... tropospheric westerly jet." -> "GCM biases in SAM persistence are correlated with biases in Southern Hemisphere jet persistent models simulating tropospheric jets which are too far equatorward (citations)."

We have changed to "Overly persistent SAM in GCMs is correlated with a common bias in the climatological jet position, whereby the simulated tropospheric jets are placed too far equatorward (citations)". Thanks.

Lines 84-86: "A possible explanation for... maintain SAM."

We have shortened it to "A possible explanation is that models with lower latitude jets exert stronger eddy-mean flow feedback to maintain SAM (Codron, 2005; Simpson and Polvani, 2016)."

Lines 96-99: "Simpson et al.... the summer season."

We have removed this sentence as it's a repetition in Section 3.3.

Line 109: "has reached beyond a sufficiently high level" -> "is sufficiently high"

Changed as suggested

Lines 110-112: "The potential role... (EERIE)."

Shortened to "Here we revisit this issue using new experiments from the Horizon Europe project European Eddy-Rich Earth System Models (EERIE)".

Lines 124-126: "In addition to the development... SST biases."

Shortened

Line 189: "To enable exploration of the response of the atmosphere to the extratropical SST ocean mesoscale features," -> "To explore the atmospheric response to extratropical ocfeatures,"

Changed

Lines 201-203: "The latter consequence...teleconnections."

Combined with the previous sentence and thus overall shortened to "The filter with a smaller L\_R at high latitudes effectively removes the smaller oceanic eddies there. However, it also removes the larger-scale tropical instability waves near the equator as when L\_R reaches its maximum. This potentially obscures the impact of targeted extratropical ocean mesoscales due to tropical-extratropical teleconnections."

Lines 277-279: "We apply...of the model."

Modified to "We first identify the latitude ( $\lambda_{max}$ ) of the maximum monthly zonally averaged 850-hPa zonal wind between 75°S and 10°S. Then, we apply a quadratic fit to the zonally averaged zonal wind at  $\lambda_{max}$  and at the two adjacent latitudes to the north and south. The latitude corresponding to the maximum value of this quadratic fit defines the position of the tropospheric westerly jet."

Lines 284: "showed a high correlation" -> "is highly correlated"

Changed

Line 302: " $cos(\lambda)$  weighting when defining the EOF in Simpson et al. (2013b)" -> " $cos(\lambda)$ -weighting (Simpson et al. 2013b)"

Changed

Lines 312-313: "While this assumption...their simulations." -> "Simpson et al. (2013b) demonstrate the validity of this assumption."

Changed

Lines 324-325: "In Simpson...of the models (Fig. 1f)."->"Following Simpson et al. (2013b), b is averaged over lags 7 to 14 days (Fig. 1f)."

Changed

Lines 330-331: "in our analysis of the model spread in simulated SAM persistence." -> "in our analysis."

Changed as suggested

Line 339-340: "it is counteracted by the negative impacts, predominantly by the surface friction, which acts to dissipate the SAM anomalies." -> "the SAM anomalies are primarily dfriction."

Modified to "While eddy momentum flux convergence primarily supports the persistence of SAM, surface friction predominantly acts to dissipate SAM anomalies."

Lines 404-406: "However, it is also possible...(e.g., CMIP6)."

Shortened to "However, it is also possible that resolution-driven improvements have plateaued within the typical grid size range of current GCMs (e.g., CMIP6)."

Lines 414-416: "For NDJ, ... jet location."

Shortened to "For NDJ, the spread of EERIE clearly shifts toward a lower  $\tau$ , closer to ERA5's  $\tau$  compared to other CMIP6 exhibiting a similar jet location."

Lines 420-423: "Still, well represented...jet location." The first part of the sentence reads well, maybe splitting would help.

Split into two sentences.

Lines 446-449: "To explore...controlled framework."

Shortened to "To explore these possibilities within a controlled framework, this section focuses on EERIE atmosphere-only sensitivity experiments with and without SST eddies (ObsSST vs. NoEddies) at two model resolutions."

Lines 476-479: "We found that...dominant mechanisms."

Shortened to "This metric correlates with  $\tau$  more strongly than b or  $[\bar{f}]_s([\bar{F}]_s)$  alone, showing a higher correlation coefficient of 0.61 and a lower p-value of 0.03 (Fig. S1d). This result points to the importance to assess the joint/net impact of the competing dominant mechanisms."

Lines 513-515: "However, the outperformance... at play."

Rewritten for clarity: "However, some CMIP6 models capture jet locations similar to EERIE, yet still perform worse for  $\tau$ , suggesting other factors are at play."

Lines 518-519: "by other factors varying in CMIP6 simulations incorporating different configurations and model systems"

Split into two sentences to improve clarity: "It is possible that the impact of resolution is outweighed by other varying factors in CMIP6, or that resolution-driven benefits have plateaued within the grid-size ranges in current CMIP6 and require more substantial resolution refinement to emerge."

Lines 551-553: "Between EERIE coupled...mesoscale features."

Shorten it to "The superior performance of the AMIP compared to coupled simulations might suggest that model skill in representing SAM persistence gains little from two-way ocean—atmosphere coupling or explicit resolving ocean mesoscale features."

Lines 559-562: "The large variability...outstanding questions." This sentence could be stronger as well as shorter.

Shorten and modified to "The large variability among ensemble members highlights the intricate mechanisms behind SAM persistence in GCMs, urging deeper investigation and alternative approaches to resolve outstanding questions regarding the atmospheric variability in the Southern Hemisphere."